# Shape-changing electrode array for minimally invasive large-scale intracranial brain activity mapping

Shiyuan Wei[1,2,7], Anqi Jiang[1,7], Hongji Sun[1], Jingjun Zhu[1,3], Shengyi Jia [1], Xiaojun Liu[1], Zheng Xu [1], Jing Zhang[1,2], Yuanyuan Shang [4], Xuefeng Fu[1], Gen Li[1], Puxin Wang[1,2], Zhiyuan Xia[5], Tianzi Jiang[6], Anyuan Cao [5] & Xiaojie Duan [1,2,3] ✉

Large-scale brain activity mapping is important for understanding the neural basis of behaviour. Electrocorticograms (ECoGs) have high spatiotemporal resolution, bandwidth, and signal quality. However, the invasiveness and surgical risks of electrode array implantation limit its application scope. We developed an ultrathin, flexible shape-changing electrode array (SCEA) for large-scale ECoG mapping with minimal invasiveness. SCEAs were inserted into cortical surfaces in compressed states through small openings in the skull or dura and fully expanded to cover large cortical areas. MRI and histological studies on rats proved the minimal invasiveness of the implantation process and the high chronic biocompatibility of the SCEAs. High-quality micro-ECoG activities mapped with SCEAs from male rodent brains during seizures and canine brains during the emergence period revealed the spatiotemporal organization of different brain states with resolution and bandwidth that cannot be achieved using existing noninvasive techniques. The biocompatibility and ability to map large-scale physiological and pathological cortical activities with high spatiotemporal resolution, bandwidth, and signal quality in a minimally invasive manner offer SCEAs as a superior tool for applications ranging from fundamental brain research to brain-machine interfaces.

Sensory, motor and cognitive operations involve the coordinated action of large neuronal populations that are widely distributed across the brain[1,2]. Understanding the neural dynamics underlying behaviour requires techniques capable of mapping brain activities at large scales and with high spatiotemporal resolution. Scalp electroencephalography (EEG), functional magnetic resonance imaging (fMRI), functional near-infrared (fNIR) imaging and magnetoencephalography (MEG) represent as powerful tools for mapping brain activity in a noninvasive manner. Scalp EEG and MEG have excellent temporal precision in the millisecond range; however, their spatial resolution is limited to centimetres and millimetres due to volume conduction[3–6]. In addition to their unfavourable spatial resolution, scalp EEG and MEG are limited in terms of their signal bandwidth[6]. The typical recording bandwidth of routine clinical EEG is -0.5–50 Hz. High-frequency oscillations (HFOs) with frequencies up to 250 Hz could be recorded by MEG or scalp EEG through specialized hardware or

[1]Department of Biomedical Engineering, College of Future Technology, Peking University, Beijing 100871, China. [2]Academy for Advanced Interdisciplinary Studies, Peking University, Beijing 100871, China. [3]National Biomedical Imaging Centre, Peking University, Beijing 100871, China. [4]Key Laboratory of Material Physics, Ministry of Education, School of Physics and Microelectronics, Zhengzhou University, Zhengzhou 450052, China. [5]School of Materials Science and Engineering, Peking University, Beijing, China. [6]Brainnetome Centre, Institute of Automation, Chinese Academy of Sciences (CAS), Beijing 100190, China. [7]These authors contributed equally: Shiyuan Wei, Anqi Jiang. ✉e-mail: xjduan@pku.edu.cn

detection algorithms[7]. However, the signal amplitude and rate are considerably lower than those of intracranial signals[8]. fMRI and fNIR generally have better spatial resolution than scalp EEG; however, their temporal resolutions are limited[4–6]. In addition, the image contrast in fMRI and fNIR relies on blood-oxygenation-level-dependent (BOLD) signals through neurovascular couplings, which indirectly reflect neural activity[5]. Intracranial electrocorticograms (ECoGs) record neural electrical activity directly from the cortex surface. ECoGs have been extensively used to identify seizure foci in patients with drug-resistant epilepsy and have become increasingly important in cognitive and brain-machine interface (BMI) studies[9–15]. The smaller distance between the signal source and detector provides ECoGs with high signal quality and spatiotemporal resolution as low as micrometres and milliseconds, respectively[4,6]. In addition to the low-frequency signals routinely observed in the EEG spectrum, ECoG can record HFOs including ripples (80–250 Hz) and fast ripples (250–500 Hz) with high fidelity[7,8]. However, ECoG recording requires a craniotomy, and an additional durotomy is needed for subdural recordings to expose the cerebral surface brain[9–14]. Craniotomy and durotomy are associated with several operative and postoperative risks, such as infections, inflammatory responses that might alter neuronal physiology and pial blood vessels, brain haematoma or oedema, and even behavioural changes[16–20]. These risks are especially high for large-area ECoG signal mapping which requires extensive craniotomy and durotomy. The invasiveness and surgery risks limit the scope of ECoG applications[15].

Here, we report the development of an ultrathin, flexible shape-changing electrode array (SCEA) for realizing large-area ECoG signal mapping with minimal invasiveness. The ultrathin electrode array, which uses a stretchable carbon nanotube (CNT) film and gold as conducting layers, was compressed from a sheet of centimetre size to a strip that was only a few millimetres wide. This strip was then inserted onto the cortical surface through a small opening in the skull for epidural recording or a small slit on the dura mater for subdural recording. Upon insertion, the SCEA strip fully deployed and changed its shape back to the ultrathin sheet with the assistance of a shape actuator, thereby forming a large-area interface with the cortical surface for brain activity mapping (Fig. 1a). With this strategy, brain-wide SCEAs were implanted epidurally into rat brains through openings in the skull as small as 2 mm by 0.8 mm, and centimetre-size SCEAs were implanted subdurally into canine brains through dural slits which were only 6 mm long. MRI and immunohistochemistry studies on the rats showed that the implantation process and chronic presence of the SCEAs caused a negligible inflammatory response or damage to the brain structure or vascular system, proving the minimal invasiveness of the implantation process and the high chronic biocompatibility of the SCEAs. SCEAs implanted through this minimally invasive procedure enabled mapping of high-quality micro-ECoG activities from rodent brains epidurally during seizure and canine brains subdurally during the emergence period. These recordings revealed large-scale spatiotemporal organization of different brain states with high resolution and bandwidth that cannot be achieved using existing noninvasive techniques. Moreover, SCEAs demonstrated capabilities to detect important electrophysiological features with minimal invasiveness, including phase-amplitude coupling (PAC) and spatial coupling, which are essential for diagnosing neurological diseases and deciphering the connectivity of neural networks. Finally, post- and intraoperative methods were developed to localize the SCEAs on brains relative to their anatomical structures. The ability to map large-area physiological and pathological cortical activities with high spatiotemporal resolution, bandwidth, and signal quality in a minimally invasive manner, as well as their high chronic biocompatibility, provide the SCEAs as a superior solution for a wide range of applications, ranging from fundamental brain research to BMIs.

## Results

### SCEAs fabrication and characterization

The SCEA was designed to achieve minimally invasive implantation of large-area ECoG electrode array through small skull/dura openings. The SCEA implantation process includes the following steps: (1) deforming an ultrathin, flexible electrode array sheet into a narrow strip in vitro; (2) inserting the SCEA strip through a small opening/slit in the skull/dura; (3) transforming the shape of the SCEA from a strip to a large-area thin sheet in vivo with the assistance of a shape actuator; and (4) withdrawing the shape actuator while ensuring that the thin electrode array remains in contact with the brain surface (Fig. 1a–c). The SCEA has a multilayer structure and was fabricated using standard photolithography and dry etching processes (Fig. 1d, Supplementary Fig. S1a). A key element of the SCEA is that the electrode array remains functional as its shape transforms between the sheet and strip, a process that is associated with extreme strains. To satisfy this requirement, we added a stretchable web-like CNT thin film on top of gold at the recording sites and proximal interconnection, where the deformation mainly located. The web-like CNT thin film was grown by chemical vapour deposition methods and consisted of strongly interconnected thin bundles of single-walled CNTs. Different from the nanometer thick gold layer, the CNT film can remain conductive after experiencing large mechanical deformations[21,22], thus preserving the functionality of the SCEAs. The CNT/Au array was placed between SU8 insulating layers with the recording sites exposed. A 4 μm thick Parylene-C layer was used as the supporting substrate.

The shape of the SCEA was transformed with a nitinol actuator. As a biocompatible shape memory alloy, nitinol exhibits a temperature-induced shape memory effect and is widely used in stents. The austenite structure of nitinol remains stable at high temperatures, and the martensite structure remains stable at low temperatures. When heated beyond the transition temperature, nitinol transforms from the martensite structure to the austenite structure and recovers its original shape and size[23]. For our SCEA application, a nitinol wire was shaped into a concave polygon (Supplementary Fig. S1b) with an austenite structure at a high temperature. The phase transition temperature was adjusted at the body temperature. After being bonded together using a water-soluble adhesive polyethylene oxide (PEO), the nitinol and CNT/Au array complex was compressed into a narrow strip at a low temperature which resulted in the martensite structure for minimally invasive implantation (Supplementary Fig. S1c). Figure 1e shows a SCEA with the nitinol actuator in its compressed state. A 20 mm × 15 mm electrode array was successfully compressed into a 3 mm wide strip. After the SCEA/nitinol was placed on the surface of an agar gel at 37 °C which was the phase transition temperature of the nitinol, the SCEA spontaneously deployed and transformed its shape back to a thin sheet with the assistance of the nitinol actuator in ~10 s (Fig. 1f). To provide a sufficient time window for SCEAs insertion in vivo, additional water-soluble PEO was applied on the nitinol/array strip. The presence of PEO introduced a delay in the start of deployment and decelerated the deployment process due to the time required for PEO dissolution. Through this approach, the deployment was effectively postponed and slowed down for several minutes (Supplementary Fig. S2, Supplementary Movie 1). After the SCEAs transformed shape to a strip and back to a sheet, due to the extreme strain and deformation experienced by the SCEAs, about 7.89 % of the channels with 1-kHz impedance below 1 MΩ ($n = 241$ from 8 devices) increased impedance beyond 1 MΩ. While for pure Au electrode array without CNT layer, a much larger fraction (40.28 %) of channels ($n = 72$ from 5 devices) increased impedance beyond 1 MΩ (Fig. 1g, h, Supplementary Fig. S3), indicating the necessity of adding CNT layer in the SCEA design. The preserved functionality of the SCEAs after the shape transformation ensures successful large-area ECoG mapping after the minimally invasive implantation.

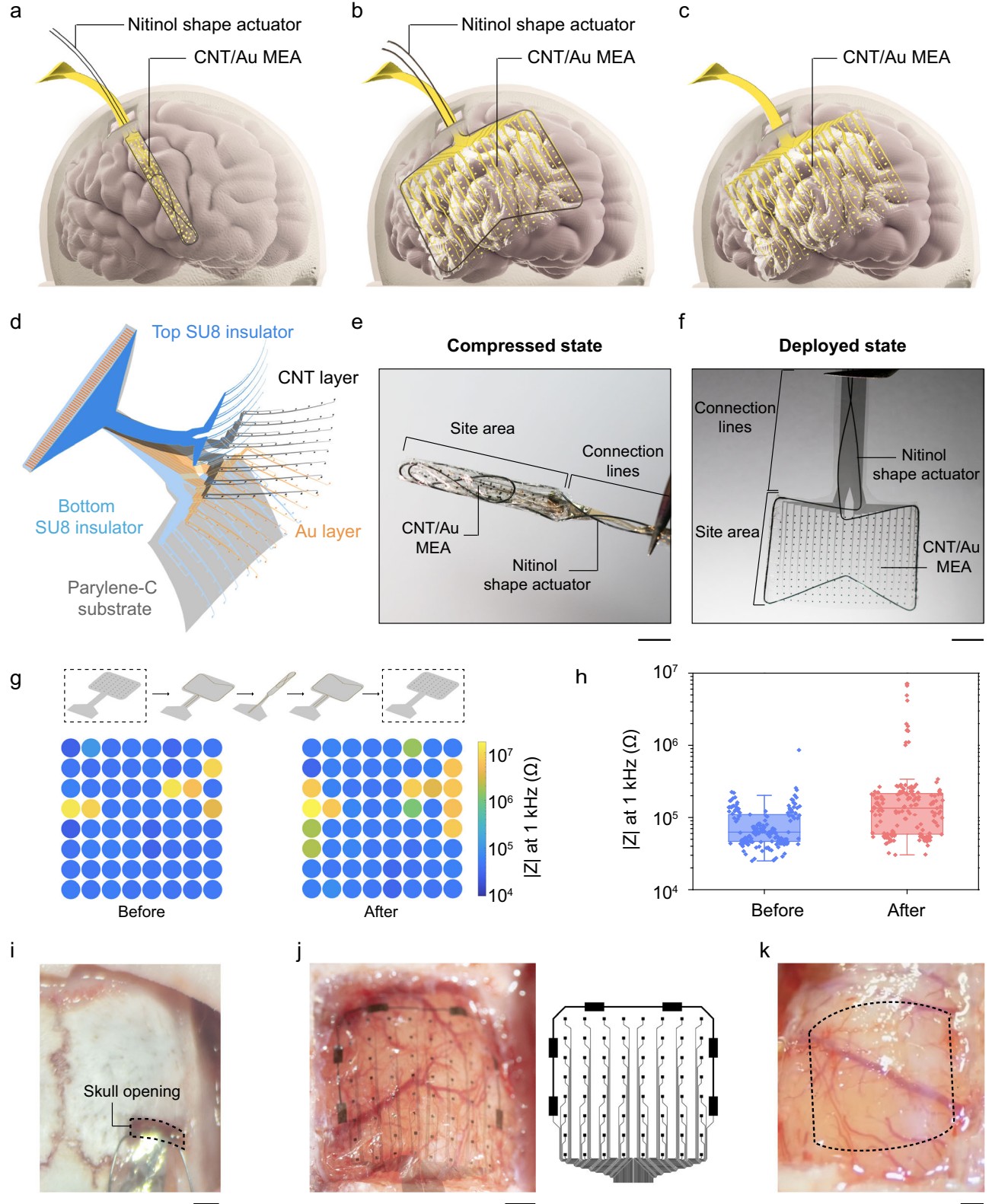

Some examples of the minimally invasive implantation of the SCEAs into rat brains are shown in Fig. 1i–k, with an entire implantation process shown in the Supplementary Movie 2 and 3, where the rat skull was thinned by drilling to enable a direct visualization. The size of the SCEAs was 6 mm × 6 mm, which covers almost the entire hemisphere of an adult rat brain. Instead of performing an extensive craniotomy, the SCEA strip was inserted into the skull through a small cranial opening that was 0.8 mm wide and 2 mm long (Fig. 1i). After washing with saline for ~40 min, the SCEAs were released from the shape actuator which was then slowly retracted through the small cranial opening. To confirm the placement of the SCEA, we performed a craniotomy after the SCEA was implanted. The SCEA transformed its shape from a strip to its fully deployed state, with all recording sites, reference sites, and interconnecting lines distributed according to the designed pattern (Fig. 1j). No bleeding was observed during any step in the implantation process, including the SCEA strip insertion,

**Fig. 1 | Design and characterization of the SCEAs. a–c** Schematics showing the design of the SCEAs for minimally invasive and large-scale intracranial brain activity mapping. **d** Expanded schematic view of the layered structure of the SCEAs. **e, f** Optical images of a SCEA in the compressed and deployed states. Scale bar, 5 mm. **g** In vitro impedance map of a 64-channel SCEA with $100 \times 100 \, \mu m^2$ active sites before and after one shape transformation cycle (compression and expansion), measured at 1 kHz. **h** Boxplots ($n = 169$ from 3 devices) showing the distribution of the 1 kHz-impedance values measured before and after one shape transformation cycle (compression and expansion). The active sites have a dimension of $100 \times 100 \, \mu m^2$. The box plots show the median and quartile range,

and the whiskers denote 1.5× the interquartile range. Individual data points are overlaid on the box plots. **i** An optical image showing the small cranial opening in a rat for SCEA strip insertion. Scale bar, 1 mm. **j** A SCEA implanted epidurally into a rat brain through minimally invasive surgery. The skull was removed after completing the SCEA implantation process for visualization. The right picture shows the designed pattern of the SCEA. Scale bar, 1 mm. **k** The cortical surface of a rat which was under a SCEA implanted through minimally invasive surgery. The dashed box shows the outline of the SCEA, which was removed for better visualization of the cortical surface. Scale bar, 1 mm. Source data are provided as a Source Data file.

deployment, or nitinol retraction (Supplementary Movie 2, 3). After the deployed SCEA was removed from the brain, the brain region under the electrode array remained intact, with no damage observed on either the brain tissue or blood vessels (Fig. 1k), demonstrating that the implantation of the SCEAs did not cause observable acute damage to the brain.

### MRI studies on rat brains after SCEAs implantation

To further evaluate the influence of the SCEA implantation process on the brain, we performed magnetic resonance imaging (MRI) studies (Fig. 2a) on rat brains at different timepoints after SCEA implantation. Figure 2b shows a representative anatomical MRI result from a rat brain with an ~6 mm × 6 mm SCEA implanted in the epidural space in the right hemisphere using the above minimally invasive surgical process. The SCEA had a mesh structure on the Parylene supporting substrate to minimize disturbances on the intracranial environment from the presence of the SCEA implant (Supplementary Fig. S4). T2-weighted images revealed no obvious abnormalities in the brain anatomical features or shape in the right hemisphere at any timepoint between 1 and 8 weeks post implantation. As a comparison, we implanted the same kind of mesh electrode array as the SCEA through conventional craniotomy method without shape transformation. These rats were more likely to experience severe brain deformations (Supplementary Fig. S5), which could be due to changes in intracranial pressure and infections during the craniotomy[16,17,19,24,25].

To evaluate whether the SCEA implantation process disrupted the vascular systems of the dura mater or brain, we conducted T2-weighted dynamic contrast-enhanced MRI (DCE-MRI). Baseline T2-weighted images were first acquired without contrast enhancement. After an intravenous bolus of the contrast agent (CA) dimeglumine gadopentetate was injected, a series of images were acquired over time. If the vascular system was disrupted, the CA would extravasate from the blood vessels and accumulate in the extravascular extracellular space, leading to elevated signal intensity in the affected tissue. Figure 2c shows a representative DCE-MRI result of a rat brain in which a mesh SCEA was implanted through minimally invasive surgery. The injection of the CA was associated with an instantaneous reduction in the signal intensity across the brain due to the flow void phenomena[26], namely, the low MRI signal intensity that occurs when fluid (including blood and cerebral spinal fluid) flows at a rapid velocity. Images acquired at subsequent timepoints showed no apparent cortical signal enhancement beneath the SCEA implant. Quantitative analyses of regions of interest (ROIs, marked by the circles in Fig. 2c) showed that at all timepoints, the signal intensity of the cortical region under the SCEA implant was comparable to that of the corresponding region in the contralateral hemisphere where no electrode array was implanted (Fig. 2c). These results indicate that there was no obvious cortical brain-blood-barrier (BBB) breach due to the SCEA implantation process, as demonstrated by the results in the acute stage, or the chronic presence of the SCEA, as demonstrated by the results in later stages. Long continuous segments of convexity meningeal enhancement were observed under the SCEA implants at 1 week postimplantation (Fig. 2c). The rats implanted with the same type of electrode array

without shape transformation through conventional craniotomy, showed similar levels of meningeal enhancements (Supplementary Fig. S6). These meningeal enhancements gradually diminished and were completely resolved at ~4 weeks postimplantation. The preservation of the brain structure and shape, as well as the limited irritation to blood vessels in the dura mater and brain, proves the minimal invasiveness of the SCEAs implantation process and the high chronic biocompatibility of the SCEAs.

### Histology studies on rat brains after SCEAs implantation

Rat brain tissues were sliced coronally at 1, 2, 4 and 8 weeks post implantation, and the slices under the SCEAs were stained using immunohistochemistry (IHC) with markers chosen to visualize the presence of neuronal nuclei (NeuN), astrocytes (GFAP), and microglia (Iba1) (Fig. 3a). The results were visualized and quantitatively compared to the results on the contralateral side, which had no implant (control), to observe any alterations in the cortical cytoarchitecture. The cortex is a laminated structure with six layers. Layer I has few to no neurons, while layer IV has the densest neuron population, which can be easily distinguished[27]. We divided the cortex into three parts in our quantitative analyses: layer I, the upper layers (UL, containing layers II, III and IV) and the lower layers (LL, containing layers V and VI). The density of neurons in the UL under the SCEA implants was comparable to that on the contralateral control side at all examined timepoints post implantation (Fig. 3b, c). The expression level of GFAP which labels reactive astrocytes elevated in layer I for the implanted side at 1 week post implantation, but showed no significant difference between the control and implanted sides for other time point or in other layers (Fig. 3b, d, e). A higher density of microglia was observed in all cortical layers under the SCEA implants than on the contralateral control side at 1 week post implantation (Fig. 3b, f–h) but not at the later timepoints. This finding indicates that in the acute stage, the SCEA implants evoked a slight inflammatory response. However, this inflammatory response diminished over time. These histological results prove the minimal invasiveness of the SCEA implantation process and the high chronic biocompatibility of the SCEAs at the cellular level.

### SCEA recordings from rat brains

The SCEAs implanted through the minimally invasive procedure can perform brain-wide cortical activity mapping with high resolution and bandwidth that cannot be achieved by noninvasive techniques such as scalp EEG, fMRI, fNIR and MEG. In one representative experiment, a 64-channel SCEA was implanted epidurally into a rat brain through minimally invasive surgery (Fig. 4a). The active sites covered a 4.6 mm × 4.6 mm region that included almost the entire left hemisphere, including the sensory, motor, and visual cortices. The potassium channel blocker 4-aminopyridine (4-AP) was applied topically on the anaesthetized rat near the upper right corner of the electrode array via a hole in the skull to induce seizures (Fig. 4a). Approximately 11 min after 4-AP administration, the SCEA recorded recurrent seizures with a duration of $113.43 \pm 88.36 \, s$ (mean ± SD) and intervals of $133.84 \pm 64.81 \, s$ (mean ± SD) that persisted for ~70 min in all cortical areas. The micro-ECoG signal in one of the SCEA channels during a

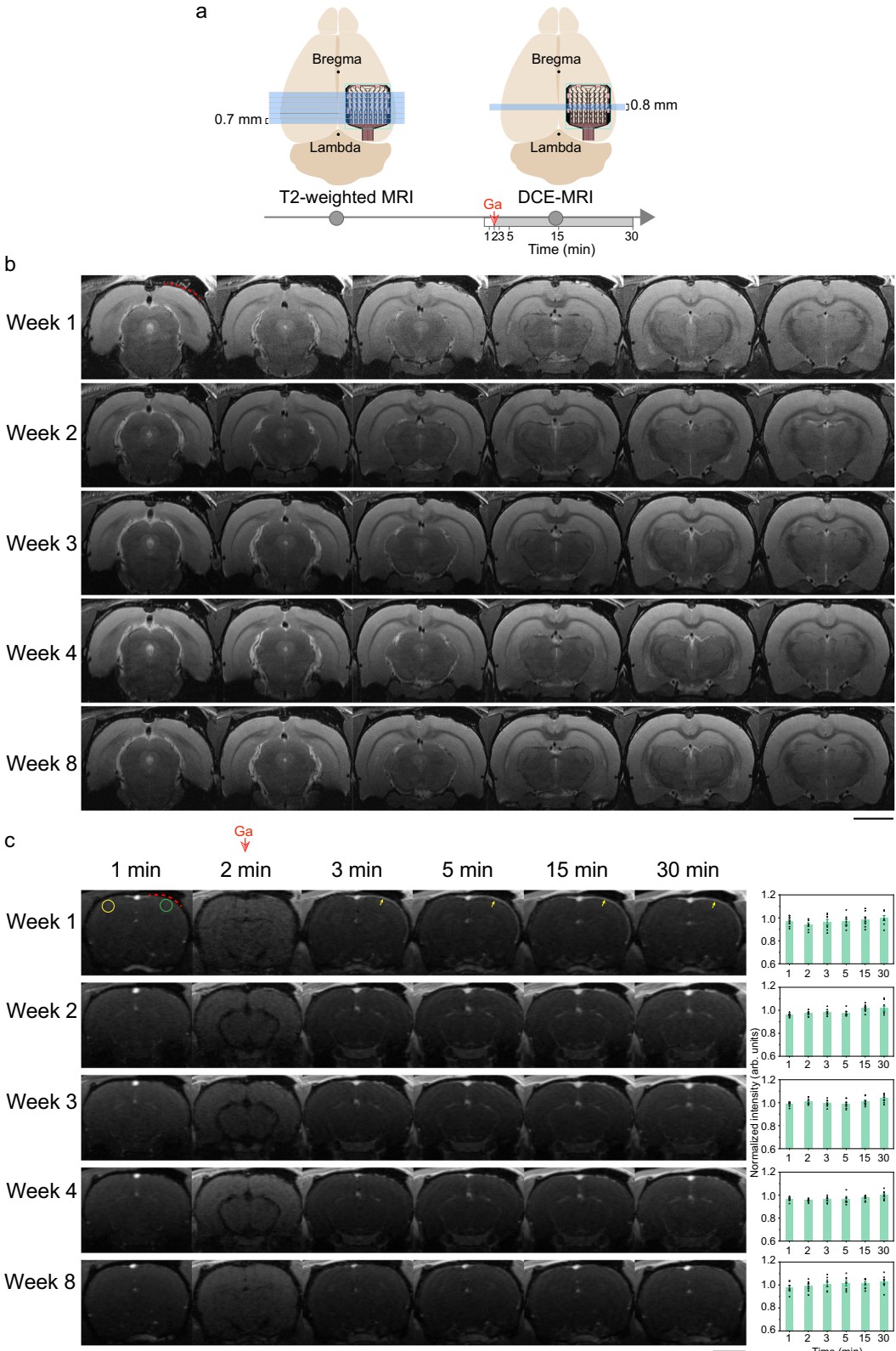

typical short electrographic seizure revealed the onset of spike and wave discharges (SWDs) after the interictal period (Fig. 4b). The power spectrogram revealed a substantial increase in the power of low-frequency components (1–30 Hz, especially the theta and alpha bands) during the ictal period (Fig. 4b). The signals from all channels, as shown in the example in Fig. 4c, revealed a spatial pattern in the epileptic spike amplitude, with larger amplitudes recorded from channels closer to the drug injection point. To better understand the spatio-temporal organization during the SWD event, micro-ECoG voltages from all 64 channels at different timepoints during an ictal spike were plotted (Fig. 4d). The results clearly demonstrate the propagation of the neural waves from regions near the drug administration site to distant regions, thus providing a method for studying the spatio-temporal evolution of brain activities.

**Fig. 2 | MRI studies on rats with implanted SCEAs. a** Schematics showing anatomical MRI and DCE-MRI measurements. **b** T2-weighted MRI images of a rat at different time points after the implantation of a mesh SCEA through minimally invasive surgery (1, 2, 3, 4, and 8 weeks postimplantation). The dotted red line indicates the position of the SCEA. Scale bar, 5 mm. **c** T2-weighted MRI images and quantification of the cortical signal enhancement during DCE-MRI measurements. The time on top of the MRI images shows the time when the MRI images were acquired during each measurement. Baseline T2 images were acquired at 1 min. The bolus injection of the CA at 2 min was associated with a reduction in signal intensity throughout the brain. The dotted red line shows the position of the SCEA which was determined from the implantation position and the geometry of the array. The yellow arrows point to locations with meningeal enhancement. The bar charts demonstrate the signal intensities of the ROIs under the SCEAs (green circle), which were normalized to those of ROIs on the collateral control side without any implant (yellow circle). $n = 8$ from 2 rats. The error bars in c represent the SEM. Scale bar, 5 mm. Source data are provided as a Source Data file.

In addition to SWD events, the SCEA successfully recorded HFOs, including high gamma activity (80–150 Hz, also referred to as gamma 2), ripples (80–250 Hz) and fast ripples (250–500 Hz) with high amplitude and high quality. These HFOs are either undetectable (fast ripples) or can only be detected with considerably lower amplitudes and rates by noninvasive scalp EEG or MEG. Figure 4e shows the high gamma and theta band signals filtered from a 2 s long segment recorded during the ictal period at the core territory of the seizure (channel #17). The data revealed a clear modulation of the high gamma signal amplitude by the phase of the theta band signal. This PAC was observed only during the ictal state and not during the interictal period (Fig. 4f). The signals recorded in regions far from the drug administration site did not show any PAC during the ictal period (Fig. 4f), which is consistent with previous findings that PACs only occur at the core territory of seizure[28]. The SCEA also detected clear discrete ripple and fast ripple signals in the pathological seizure state (Fig. 4g). The fast ripples manifested as 'islands' in the power spectrogram and could be distinguished from 'false' activity arising from applying a high-pass filter to sharp events[28].

### Subdural recordings from canine brains with SCEAs

The strategy of using the SCEAs for minimally invasive intracranial brain activity mapping can also be applied subdurally in large animal models. The larger thickness of the dura mater in large animals causes more severe neural signal attenuation than in rodents. Hence, compared to epidural recordings, subdural recordings are more desirable for large animal models. For large-scale subdural cortical activity mappings from large animal models, an extensive durotomy is typically required which is associated with even higher risks than the craniotomy because the removal of a large-size dura mater leads to changes in intracranial pressure, increased risk of cerebrospinal fluid (CSF) leakage, and potential infections[29–32]. We fabricated SCEAs for and implanted them to beagle dogs subdurally using a minimally invasive implantation method without the need for durotomy. After the skull was removed, the SCEAs in the compressed state were inserted into the subdural space through a ~6 mm long dural slit without removing any dura mater (Fig. 5a). Pictures of a beagle brain surface during the deployment of a SCEA under the dura are included as Supplementary Fig. S7. The SCEAs expanded spontaneously from strips to their original sheet shape under the dura mater in ~10 s upon the insertion. Figure 5b shows the subdural implantation of a 20 mm × 15 mm 256-channel SCEA into a beagle dog before and after the nitinol shape actuator was retracted. We performed four subdural implantations in two beagle dog brains. There were some cases where the craniotomy caused some bleeding. But no bleeding was observed during the SCEAs insertion, deployment or shape actuator withdrawal processes. For all implantation, the vessels on the cortex surface and the brain tissues remained intact. In addition, the small slit in the dura prevented outward swelling and shift of the brain, which are always observed during large-area durotomy. These findings highlight the safety and minimal invasiveness of the subdural SCEA implantation procedure.

The SCEAs implanted subdurally with the minimally invasive procedure were capable of mapping neural activities across wide cortical regions with high spatiotemporal resolution and bandwidth in large animal models. In one representative experiment, a 20 mm × 15 mm 64-channel SCEA was implanted subdurally into the occipital and temporal lobes of a beagle dog, covering the ectolateralis posterior (QP), suprasylvian gyrus (SSM) and ectosylvian gyrus (EM) regions, using the above minimally invasive surgical procedure (Fig. 5c). The SCEA recorded high-quality micro-ECoG signals under the anaesthesia which was maintained at the minimal alveolar concentration level with inhaled isoflurane, and during an emergence period when the animal gradually returned to a conscious state (Fig. 5c)[33]. The mean spectrum across the 64 channels revealed higher levels of alpha/beta (~10–30 Hz) and gamma 1 band (~30–80 Hz) power in the emergence state than in the anaesthesia maintenance period (Fig. 5d). The power density spectrum of an individual channel confirmed the increase in alpha/beta and gamma 1 oscillation power during the emergence period (Fig. 5e). In addition, the SCEA provided spatially resolved patterns of the neural state change, as shown by the power spectral density (PSD) maps in the beta and gamma 1 bands across different channels (Fig. 5f).

With high spatiotemporal resolution and bandwidth, subdural recordings with the SCEAs enabled studies on coupling between distributed brain regions under different brain states in a minimally invasive manner. The coupling between different recording sites was assessed using the coherence, a frequency domain measure of linear association that is widely used in neuroscience studies[34]. As seen in the computed channel-to-channel coherence maps and distributions in Fig. 5g, h, the gamma band, including the gamma 1 band of 30–80 Hz and the gamma 2 band of 80–150 Hz, showed the lowest channel-to-channel coherence, while the delta band showed the highest coherence among all bands in both the anaesthesia and emergence states. This phenomenon occurred partially because low-frequency signals can couple over considerably longer distances, while coupling of high-frequency signals is more spatially limited[35]. Another possible cause of the low gamma coherence is the low level of consciousness caused by general anaesthesia. It has been reported that general anaesthesia reduces functional interactions in various cortical and subcortical regions, leading to alterations and interruptions in cortical integration during information processing, which is encoded by gamma activity synchronization[36,37]. Figure 5i, j show maps of the statistical significance of the coherence difference between the anaesthesia and emergence states and the distribution of the significance across channel pairs. The delta and beta oscillations had the largest number of electrode pairs with a coherence increase (44.0% and 46.2%) during the transition from unconsciousness to the emergence state (Fig. 5j). Despite the increased power in the gamma bands, the emergence state demonstrated a lower coherence level, with ~55.7% and 37.0% of the pairs exhibiting significant decreases for the gamma 1 and gamma 2 bands, respectively. These results demonstrate that the SCEAs are capable of tracking spatiotemporal neural dynamics with high resolution and bandwidth for investigating brain network integration and segregation.

### SCEA localization

Localization of electrode sites on the brain surface is required to relate electrophysiological and anatomical features, which is critical for both fundamental neuroscience research and clinical diagnoses and

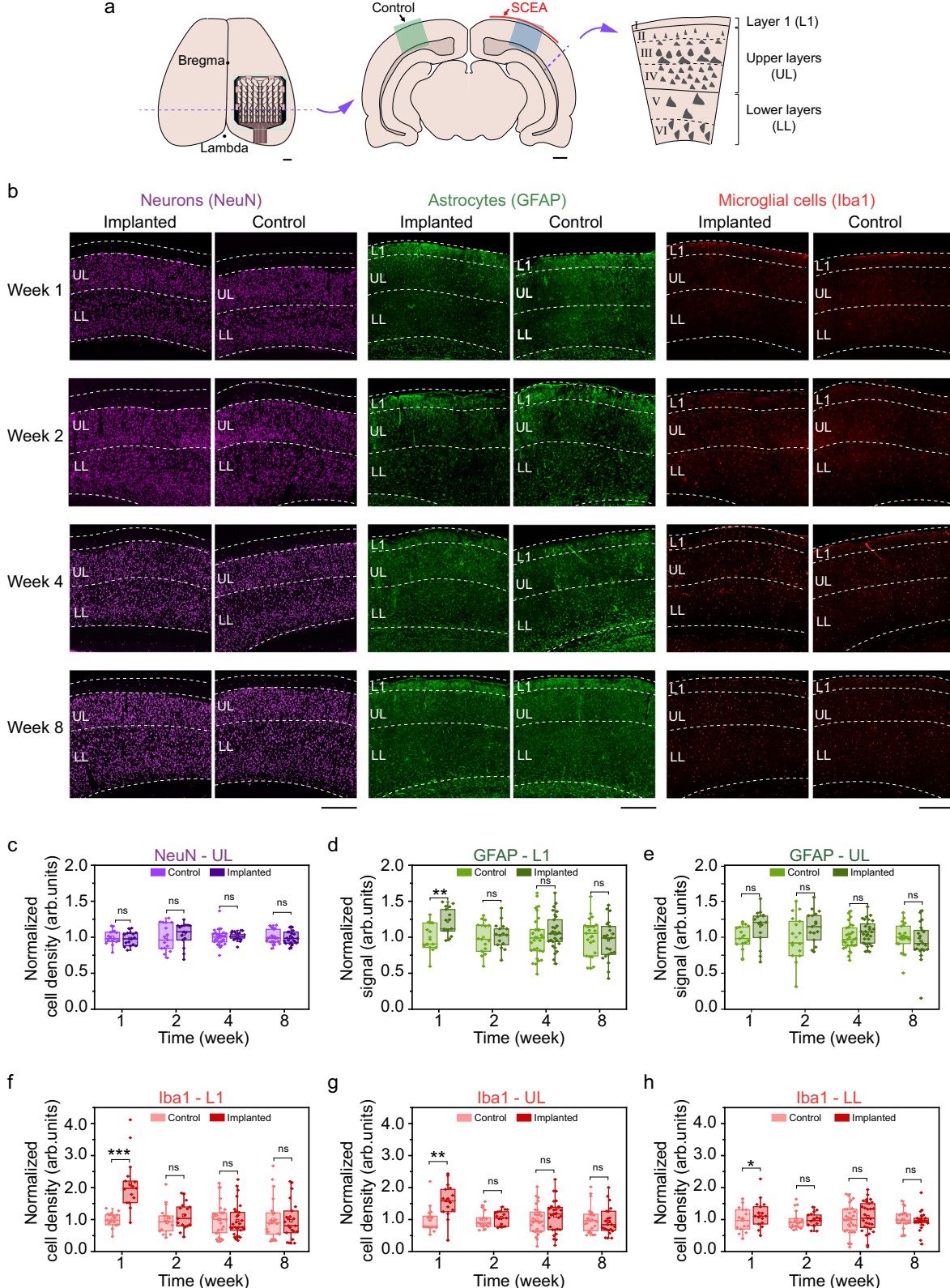

**c** NeuN - UL **d** GFAP - L1 **e** GFAP - UL

**f** Iba1 - L1 **g** Iba1 - UL **h** Iba1 - LL

treatment of neurological diseases. For conventional ECoG electrode arrays implanted through craniotomy or durotomy, intraoperative photographs provide a simple method for localizing the implants on the cortical surface[38]. In contrast to beagle dogs, which have semi-transparent dura mater, many other large animal models such as nonhuman primates or humans have opaque dura mater. The absence of craniotomy or durotomy during the SCEA implantation process

prevents intraoperative visualizations and photographs of the electrode contacts and cortical surfaces. The nanometre thick metal lines on the SCEAs prevent the use of postoperative computed tomography (CT) imaging for electrode localization[38].

To achieve postoperative SCEA localization on the brain surface, we developed an MRI-based method with a 100 nm thick nickel layer as a contrast enhancer. The nickel layer was placed between the bottom

**Fig. 3 | IHC studies on rats with implanted SCEAs. a** Schematics showing the IHC assessment. The SCEAs were implanted epidurally through minimally invasive surgery in the right hemisphere of the rat brains. Contralateral sides with no implants served as controls. The blue and green blocks illustrate the ROIs in the quantitative analyses. In the quantitative analyses, the cortex was divided into three parts: L1, the UL, and the LL. **b** Example immunofluorescence images of a rat cortex under the SCEA (implanted) and on the contralateral control side (control) at different time points after SCEAs implantation. Scale bar, 500 µm. **c** Normalized cell density of NeuN-labelled neurons in the upper layers. **d, e** Normalized signal of GFAP-labelled astrocytes in layer I (**d**) and the upper layers (**e**). **f–h** Normalized cell density of Iba1-labelled microglial cells in layer I (**f**) the upper layers (**g**) and the

lower layers (**h**). The cell density (for the neurons and microglia) and signal (for the astrocytes) on the implanted sides were normalized to those on the control side (average values from all animal samples at each time point). Individual data points are overlaid on the box plots. The box plots show the median and quartile range, and the whiskers denote 1.5× the interquartile range. $n = 16$ from 4 rats for weeks 1 and 2 data, $n = 29$ from 4 rats for week 4 data, $n = 23$ from 6 rats for week 8 data. For the data that could be represented by a normal distribution, two-sided paired t tests were used. Otherwise, Wilcoxon signed-rank tests were used in the significance analysis. ns, $p > 0.05$, *$p < 0.05$, **$p < 0.01$, ***$p < 0.001$. $p = 0.0029$ (week 1, **d**), $p = 0.0000$ (week 1, **f**), $p = 0.0052$ (week 1, **g**), $p = 0.0496$ (week 1, **h**). Source data are provided as a Source Data file.

SU8 layer and the Parylene substrate with the same pattern as the CNT/ Au layers (Ni-SCEA, Supplementary Fig. S8). As shown in the MRI images of a beagle dog brain with a subdurally implanted Ni-SCEA (Fig. 6a, b), the nickel layer significantly increases the image contrast of the SCEA in MRI scans. The serial MRI images were used to obtain 3D reconstructed MRI images of both the brain and the electrode array (Fig. 6b). Figure 6c shows three sections cut along the cortical surface in the reconstructed 3D images. Due to the curvilinear nature of the cortical surface and the conformal interface between the SCEA and the brain, one planar section could not cover the entire array and instead covered a portion of the array. By registering the MRI population template[39] to the subject's native space (Fig. 6d), the position of the SCEA was determined relative to anatomical structures according to the loaded atlas.

An alternative intraoperative localization method was developed for the SCEAs which utilized the commercial Brainsight™ neuronavigation (Vet Robot) system, which is a frameless imaging-guided stereotaxic system[40]. This method is especially suitable for acute studies where the SCEAs are not left on the animals' brains after the measurements. Briefly, a 3D reconstructed image of the skull and brain was obtained from coregistration of CT and MRI images (Fig. 6e). Based on the hundreds of homologous fiducials on the skull, the animal was registered to the 3D CT/MRI images by creating a common coordinate system between the physical space and the virtual reconstruction space. Any position on the animal's head could be targeted physically and defined in this common coordinate system with a 6-axis robotic arm. For subdural implantation of the SCEAs in large animal models with opaque dura mater, the interconnection lines at the dura slit exit were visible. By marking the coordinates of three points on this part of interconnection lines (Fig. 6f), the coordinates of the recording sites, i.e., their exact localization in the brain, were defined according to the layout of the electrode design.

## Discussion

Compared to other techniques for brain-wide neural activity mapping, intracranial ECoG recording possesses higher spatiotemporal resolution, signal quality, and bandwidth. However, the invasiveness, intraoperative risks and postoperative complications associated with the implantation of the ECoG electrode array limit their applications. These issues are even worse for large-scale ECoG mapping which requires mass craniotomy and durotomy. Here we realized the implantation of large-area ECoG array and mapping of high-quality cortical activity both epidurally and subdurally in a minimally invasive manner using the strategy of SCEAs. With the assistance of the nitinol shape actuator, the SCEAs fully expanded under the skull or dura mater, and formed a large-area interface with the brain. Structural MRI studies on rats indicated no change of the brain structure or shape after SCEAs implantation. DCE-MRI studies observed no obvious cortical BBB breach after the SCEAs implantation either. Segments of convexity meningeal enhancement were observed under the SCEA implants at 1 week postimplantation. Similar levels of meningeal enhancements were also observed on rat brains implanted with the same type of electrode array through craniotomy and without shape

transformation. The meningeal enhancements observed here gradually diminished and were completely resolved at ~4 weeks postimplantation. Extensive linear meningeal enhancements, which reflect vascular abnormalities in the dura mater, have been reported to arise due to various factors, including vasocongestion, increased vascularity or angiogenesis in the dura, which is usually leakier due to immaturity, interstitial oedema, and dural hypertrophy[26,41,42]. The fact that meningeal enhancements also occurred when the electrode array was implanted through craniotomy suggests that these enhancements were not caused by disruptions in the dura mater or meningeal blood vessels due to the electrode deployment or nitinol actuator withdrawal processes; instead, they might be from the irritation of the meningeal blood vessels in dura mater by the presence of the electrode implants. This is consistent with a previous study which observed the formation of microhaematomas and vasculature in the dura mater of rat brains after regular epidural ECoG implantation through craniotomy, with the haematomas completely resolved and the vasculature stabilized ~24 days after implantation[42]. Our MRI and histology studies on rats proved that the implantation process, including the SCEA strips insertion, deployment, and shape actuator retraction, was minimally invasive and the SCEAs had excellent chronic biocompatibility.

ECoG electrode array has been inserted through a small cranial slit and slid onto the epidural cortical surface in mice for functional connectivity mapping[17]. The direct insertion method can also avoid the oedema and deformation in the brain associated with mass craniotomy. Compared to this direct insertion method, our SCEA strategy have two main advantages. Firstly, it allows for the expansion of the electrode array in the direction perpendicular to the insertion pathway, thus results in larger cortical coverage of the ECoG mapping. Secondly, the SCEA strategy allows for the use of much thinner electrode array than the direct insertion method, for which the electrode array must have enough thickness and rigidity to enable the insertion and sliding. Thinner electrode array forms more conformal interfacing with the brain surface, which improves both the signal quality and stability because random movements of the implants in the brain are prevented. In addition, it was reported that the ultraconformal interfacing could even record action potentials from superficial cortical neurons[9]. The SCEA strategy doesn't rely on the mechanical rigidity of the electrode array for insertion and deployment, thus having no limitation on the thickness of the electrode array. It is noted that the deployed SCEAs could be safely removed from the intracranial space through the small cranial opening and remain intact, owing to their compliant nature and excellent mechanical strength, as shown in Supplementary Movie 3. This provides great convenience for acute applications of the SCEAs.

The SCEAs implanted through the minimal invasive procedure recorded high quality micro-ECoG signals both epidurally from rat brains and subdurally from canine brains, including HFOs with frequencies of up to 500 Hz. The high-quality recording detected important electrophysiological features, including PAC and spatial coupling which are essential for diagnosing neurological diseases and deciphering the connectivity of neural networks. Previously, many of

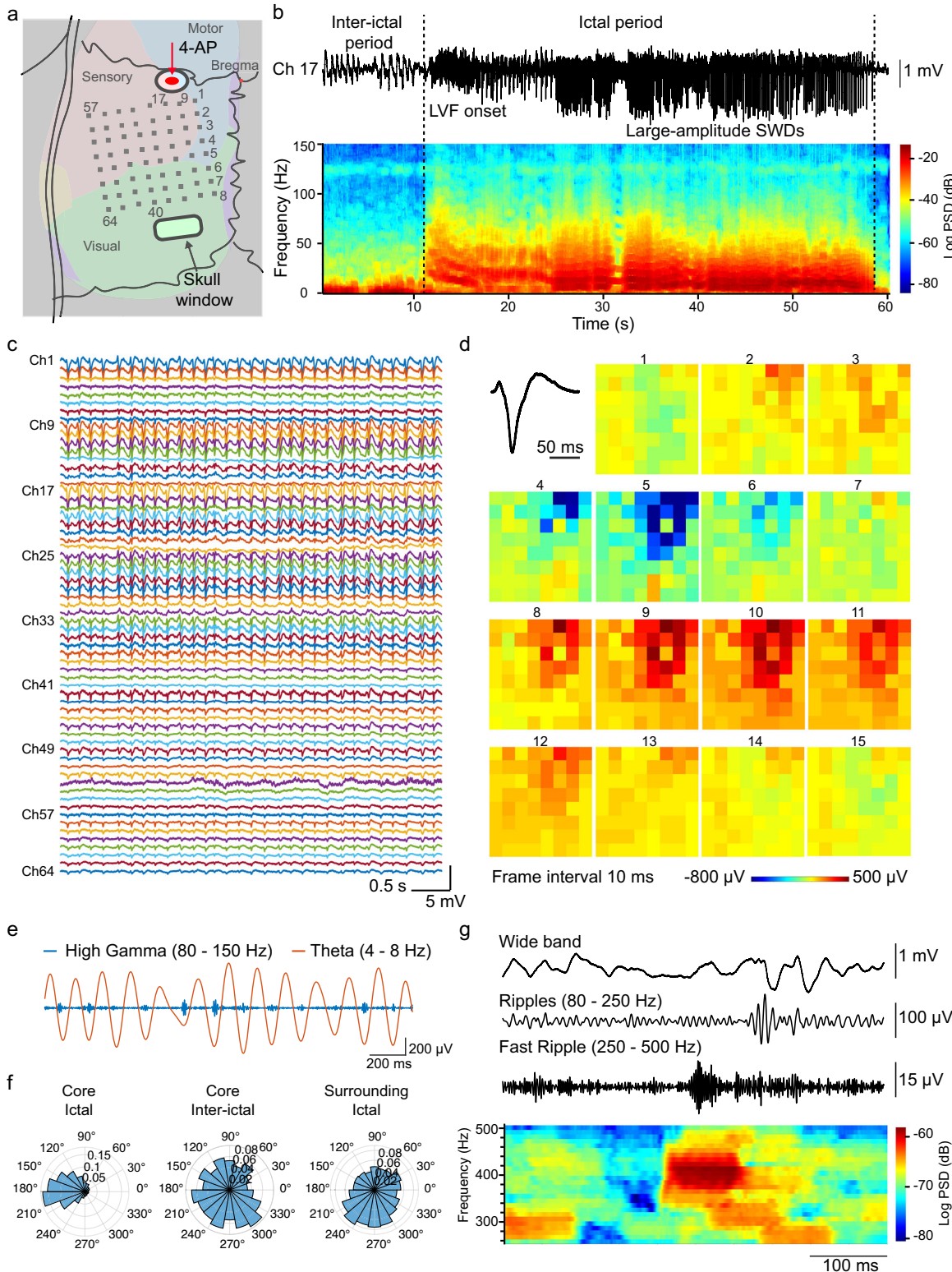

these electrophysiological features could only be observed using invasive techniques. Here, we show that with the SCEA strategy, these features can be reliably detected in a minimally invasive manner. We believe that the SCEAs' capability of large-scale mapping physiological or pathological cortical activities with high spatiotemporal resolution, signal bandwidth, and signal quality in a minimally invasive manner could offer unique opportunities in both fundamental neuroscience

studies and high-precision diagnoses and treatment of various neurological diseases.

## Methods

### Fabrication and characterization of SCEAs

The fabrication steps of a SCEA were as follows (Supplementary Fig. S1): (1) A 1.5-μm-thick negative photoresist (SU8-2002, MicroChem

**Fig. 4 | A representative SCEA recording in a rat. a** Illustration of SCEA recordings in rats. The SCEAs were implanted epidurally through minimally invasive surgery. 4-AP was applied topically through the cranial hole marked by the oval. The recording channels were labelled, as shown by the numbers. **b** Micro-ECoG signal and its spectrogram recorded in a representative channel during a short electrographic seizure recorded in an experiment. The dotted lines indicate the onset and termination of ictal activity. The seizure activity initially manifested as low-voltage fast-onset (LVF) patterns. Then the frequency of the ictal spikes decreased as the spike amplitude increased over time, with the activity changing to large-amplitude SWDs. **c** A segment of the ictal activity recorded by the SCEA. **d** Movie frames show varied spatial-temporal micro-ECoG voltage patterns during a representative SWD (black trace, upper left corner) period. The signals were bandpass filtered from 1 to 50 Hz. The frame interval is 10 ms, and the colour scale is saturated at −800 μV and 500 μV to improve the visual display. The data are anatomically oriented according to the pattern in **a**, with the upper right most site marked as channel 1 and the lower left most site marked as channel 64. **e** An example segment of the recorded data, showing the coupling between the amplitude of the high gamma band (blue trace) and the phase of the theta band (orange trace) during the ictal period. **f** Polar plots showing the coupling between the amplitude of the high gamma band and the phase of theta oscillations in the core area during the ictal and interictal periods and the surrounding areas during the ictal period. **g** Examples of HFOs recorded by the SCEA during the ictal period. No filtering was performed for the wide band signal. The spectrogram at the bottom reveals the elevated power of the 250–500 Hz component, which is consistent with the presence of fast ripple HFOs. Source data are provided as a Source Data file.

Corp., USA) was patterned using standard photolithography as the top passivation layer on a silicon wafer with nickel sacrificial layer. (2) Web-like CNT film, prepared through floating catalyst chemical vapour deposition and a solvent-induced condensation process as previously reported[21], was transferred onto the patterned wafer. (3) Metal layer consisting of Cr/Au (8 nm/60 nm) was formed using photolithography (ARP-5350, MicroChem Corp., USA) and physical vapour deposition (PVD75, Kurt J. Lesker Co., USA). This metal layer was used as active sites, interconnection and connection pads, as well as the protective mask for following dry etching of the CNT film. (4) CNT film was patterned by an oxygen-based inductively coupled plasma-reactive ion etching (ICP-RIE) process (300 W, 80 sccm $O_2$ for 2 min, Minilock III ICP, Trion Co., USA). The exposed active recording sites were 100 μm and 200 μm large squares for applications on rodent and canine models, respectively. (5) A 0.5-μm-thick SU8 layer (SU8-2000.5, MicroChem Corp., USA) was patterned using standard photolithography as the bottom passivation layer. Samples were subsequently baked at 190 °C for 1 h for further curing and fusion of the two SU-8 layers. (6) Parylene C was deposited to a thickness of 4 μm as the supporting substrate (PDS 2010, Specialty Coating Systems Inc., USA). (7) For mesh electrode array fabrication, Parylene was etched by ICP-RIE (300 W, 80 sccm $O_2$ for 10 min, Minilock III ICP, Trion Co., USA) with a 120-nm-thick aluminum layer as protective mask. (8) By immersing the samples in 1 M $FeCl_3$ solution, the nickel layer and the aluminum mask for the mesh electrode array was etched away. The CNT/Au MEA was released from the wafers. (9) The CNT/Au MEA was electrically connected to a custom PCB through a heat seal connector (HSC) film (Chuangli Commercial Co., China). (10) A nitinol wire with a diameter of 100 μm or 250 μm (Ni: 55.61 wt%, Jiangyin Lumenous Peiertech Corp., China) was used to fabricate the shape actuator. A custom metal mould consisting of seven screws on a plate was used to shape the nitinol wire. After wrapping the nitinol wire tightly (Supplementary Fig. S1b), the screws were screwed onto the metal plate. Following heat treatment at 480 °C for 40 min, a quenching process in cold water was carried out. As a result, the nitinol wire was shaped into a rectangle with a concave edge, with the transition temperature adjusted at the body temperature for SCEA application. (11) The shaped nitinol actuator was bonded onto the backside of the CNT/Au electrode array using water-soluble polymer adhesive polyethylene oxide (PEO, 15 wt%). (12) The SCEA/nitinol complex was transferred onto the surface of dry ice, resulting in the phase transition from the austenite to martensite phase which was more mechanical flexible. (13) The SCEA/nitinol complex was mechanically compressed into a strip in the dry ice environment. PEO was applied onto the strip to achieve a larger time window for the insertion.

Electrochemical impedance spectroscopy (EIS) measurements were carried out on a CHI660e electrochemical workstation (CHI660e, CH Instruments Inc., China). The test was performed in 1 × PBS with pH 7.4 at room temperature, using three-electrode configuration with an Ag/AgCl electrode as the reference electrode and a Pt wire as the counter electrode.

## Implantation of SCEAs into rats

Animal experiments were conducted complying with Beijing Administration Rules of Laboratory Animals and the national standards of Laboratory Animal Requirements of Environment and Housing Facilities (GB 14925-2010). The rodent experiments were approved by the Institutional Animal Care and Use Committees of Peking University (#COE-DuanXJ-1). SCEAs were implanted to rats epidurally through the minimally invasive procedure. Male Sprague Dawley rats (Charles River Laboratories Inc., China) weighing 300–400 g were used. The animals were pretreated with mannitol (2 g/kg at 20 wt%) through caudal vein injection 1 h prior to surgery in order to reduce intracranial pressure. The rats were anesthetized initially with 5% (0.6 L/min) isoflurane and then the anesthesia was maintained with 0.5–1.5% isoflurane during the surgery in a lab standard stereotaxic instrument (51600, Stoelting Co., USA). A small skull opening (2 mm × 0.8 mm) was made on the right skull, with the lower left corner of the opening being 1 mm anterior and 3 mm lateral from the lambda (Fig. 1i), and from which a SCEA strip in its compressed state was slowly inserted into the epidural space. Since the nitinol actuator was attached to and remained on the backside of the electrode array, by appropriately orienting the strip during the implantation, it can be ensured that the active site side faces the brain tissue. Before insertion, the front part of the head was lowered to achieve an inclined plane at an angle of about 30° relative to the horizontal plane of the ear rods. This way the insertion direction was maintained parallel to the skull plane to avoid compression of the brain. After the insertion, the rat head was adjusted to the horizontal plane. Saline was repeatedly injected to dissolve PEO on the SCEA strip. About 40 min after the insertion and washing, the nitinol shape actuator and the CNT/Au electrode array were fully separated, and then the shape actuator was slowly retracted through the small cranial opening, leaving the SCEA adhered to the brain epidural surface. The cranial window was sealed with noncytotoxic silicone elastomer (Kwik-Sil, World Precision Instruments Inc., USA), followed by application of photosensitive resin.

## Implantation of the electrode array through craniotomy

CNT/Au electrode array with the same design, size and fabrication method as the mesh SCEAs were used after being released from the silicon wafers without any shape change. For an implantation, a 6.5 mm × 6.5 mm cranial window was opened by drilling in the right hemisphere of a rat brain, with the lower left corner of the opening being 1 mm anterior and 1 mm lateral from the lambda. After removing the skull, a CNT/Au electrode array sheet was directly placed on top of the dura mater. The removed cranial segment was put back to its original position. Finally, the cranial gap produced by drilling was sealed with noncytotoxic silicone elastomer (Kwik-Sil, World Precision Instruments Inc., USA), followed by application of photosensitive resin.

## MRI studies

All MRI experiments on rats were performed in a Bruker 9.4 T scanner (Bruker BioSpin 94/20USR MRI) with Bruker's 86 mm volume coil for

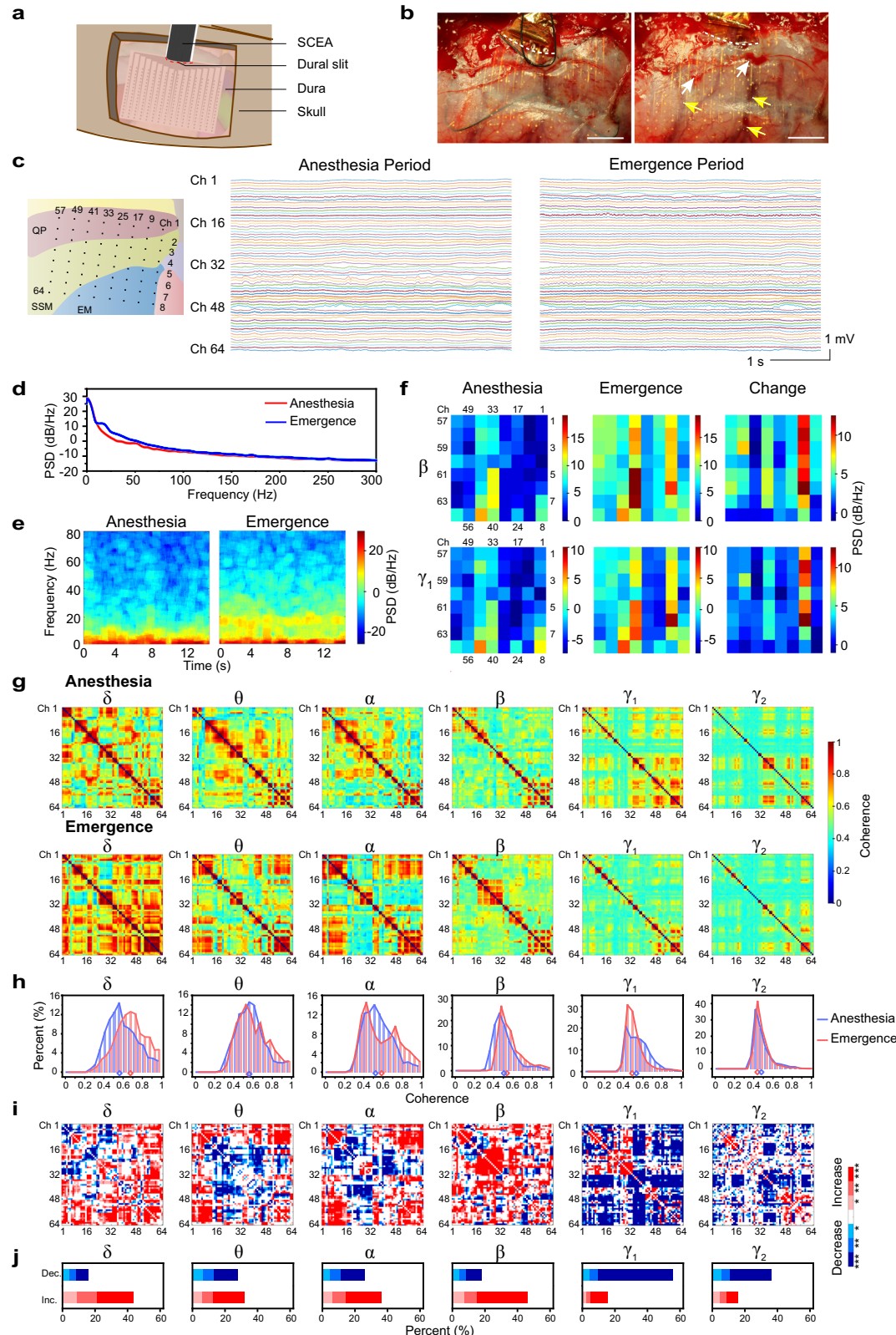

transmission and a four-channel rat head surface coil for receiving (ParaVision Version 6.0.1 for MRI acquisitions). In order to minimize the disturbance on the MRI signals, the electrode implants were not connected to the HSC or PCB. The interconnection line part was tethered onto the skull and sealed in the photosensitive resin. MRI studies were conducted at 1, 2, 3, 4, and 8 weeks after the electrode implantation. For MRI scanning, the animals were firstly anesthetized

with 5% isoflurane and then were maintained under anesthesia with 0.5% isoflurane in oxygen delivered via a nose cone. Animal temperature, respiration and blood oxygen saturation were all monitored and within normal ranges (Model 1025, SA Instruments, USA).

T2-weighted anatomical images were acquired with parameters as follows: repetition time (TR) /echo time (TE) = 2200/33 ms, RARE factor = 8, field-of-view (FOV) = 25 × 23 mm², matrix size = 512 × 512, and

**Fig. 5 | Subdural SCEA implantation and recording in canine models.**
**a** Schematic showing the minimally invasive subdural implantation of the SCEAs in large animal models. **b** Optical images acquired during subdural implantation of a SCEA into the brain of a beagle dog before and after withdrawal of the nitinol actuator. The dura slit is marked by the white dotted line. The craniotomy caused some bleeding in the skull, leading to the formation of several hemorrhagic spots on the surface of the dura (white arrows), while the blood vessels underneath the dura displayed a much darker red colour (yellow arrow). The vessels under the dura remained intact, and no bleeding was observed during the SCEA insertion, deployment or shape actuator withdrawal processes. Scale bar, 5 mm. **c** Signals recorded from a beagle dog during the anaesthesia and emergence periods (6 min after discontinuing the administration of isoflurane). The left schematic shows the location of the SCEA. QP ectolateralis posterior; SSM suprasylvian gyrus, EM ectosylvian gyrus. **d** The averaged power spectrum during the anaesthesia and emergence periods. **e** Spectrograms of signals recorded by a representative channel in the anaesthesia and emergence periods. **f** Spatial power spectral density (PSD) patterns in the beta and gamma 1 band during the anaesthesia and emergence periods, as well as the PSD difference between these two states. **g** Channel-to-channel coherence map of different bands during the anaesthesia and emergence periods. Note that the colour bar scale is equalized across the matrices for ease of comparison. **h** Histograms showing the distributions of the coherence values in the anaesthesia and emergence states. The rhombuses at the bottom indicate the medians of the distributions. **i** Statistical significance of the coherence difference between the anaesthesia and emergence states. A significant increase is indicated by red pixels, and a significant decrease is indicated by blue pixels. *$p < 0.05$, **$p < 0.01$, ***$p < 0.001$, two-sided Wilcoxon signed-rank test. **j** Histograms showing the distribution of the significant difference in (**i**). Source data are provided as a Source Data file.

slice thickness = 0.7 mm. Six contiguous coronal slices under the implant were scanned for each animal at each time point (Fig. 2a). DCE-MRI images were acquired using FLASH sequence with parameters as follows: TR/TE = 15.625/1.7 ms, flip angle = 18°, FOV = $25 \times 23$ mm$^2$, matrix size = $128 \times 128$, and slice thickness = 0.8 mm. Baseline images were firstly acquired without contrast agent (CA). Four contiguous coronal slices under the implanted electrode array were scanned for each animal at each time point (Fig. 2a), with one of the images presented in Fig. 2c. Dimeglumine gadopentetate (Beijing Beilu Pharmaceutical Co., Ltd., China) was bolus injected at a dose of 0.5 mmol/kg 2 min after the baseline image scanning. Acquisition of images was continuously performed for 30 min. To quantify the enhancement of cortical signal intensity, two circles with a diameter of 1.5 mm were selected underneath the implants and at contralateral position as regions of interest (ROIs), The center of these circles was 4 mm lateral from the longitudinal fissure. The ratio of the mean gray values in the two circles, defined as normalized intensity as shown in Fig. 2c, were calculated at each time point as the indicator for cortical BBB breach.

## Immunohistochemistry studies

The SCEAs were implanted epidurally in the right hemisphere of rats used for immunohistochemistry measurement using the minimally invasive surgery. 1, 2, 4, and 8 weeks after the SCEAs implantation, rats were anesthetized and transcardially perfused with 100–150 mL PBS, followed by 200–300 mL 4% paraformaldehyde (PFA) in PBS. After being immersed in 4% PFA solution for 24 h at room temperature for the purpose of post-fixation, the brain was immersed successively in 15% (w/v) and 30% (w/v) sucrose solution for dehydration until it settled to the bottom of the solution. The brain was segmented coronally to remove extra brain tissue with only ~5 mm thick section under the implants left (starting from 1 mm to 6 mm anterior lambda). The tissue sample was cryoprotected using the optimal cutting temperature (OCT) compound (Tissue-Tek, USA) and frozen at −80 °C to stiffen the OCT compound. Frozen tissues were sliced coronally into 8 μm thick sections using a cryostat machine (Leica CM3050 S, Germany), mounted on glass slides and stored at -20 °C.

0.3% (v/v) TritonX-100 in PBST (0.1% Tween-20 in 1 × PBS) was applied to the brain slices for 20 min. After being rinsed with PBST, the slices were blocked with 5% (w/v) bull serum albumin (BSA) at room temperature for 2 h and washed by PBST solution. Then slices were incubated for 14–16 h at 4 °C with primary antibodies: chicken anti-glial fibrillary acidic protein (GFAP) (targeting astrocytes, 1:1000, Abcam #ab4674, USA), goat anti-ionized calcium binding adaptor molecule 1 (Iba 1) (targeting microglia, 1:500, Abcam #ab5076, USA), and rabbit anti-neuronal nuclear (NeuN) (targeting nuclei of neurons, 1:1000, Abcam #ab177487, USA). After thoroughly washed with PBST solution for 5 times, the slices were incubated in a secondary antibody solution, which include: Alexa Fluor 647 donkey anti-rabbit (1:500, Abcam #ab150075, USA); Alexa Fluor 568 donkey anti-goat (1:500, Abcam #A11057, USA); Alexa Fluor 488 donkey anti-chicken (1:500, Jackson

Immunoresearch #703-545-155, USA). This incubation was kept away from light and lasted for 2 h at room temperature, followed by washing up with PBST solution for 5 times. Afterwards, the brain slices were incubated at dark with 4′,6-diamidino-2-phenylindole (DAPI, Sigma-Aldrich, USA) for 15–30 min, and followed by rinsing with PBS solution and DI water subsequently. The brain slices were sealed with antifading mounting medium (Solarbio, China) and coverslips and stored at -20 °C without exposure to light, allowing for 7 days of clearance before imaging.

Confocal imaging was performed on a DIVE system (TCS SP8 DIVE, Leica, Germany). Fluorescence images were taken at multiple focal depth of a tissue slice to ensure the image quality. Images recorded from the same brain slice but the opposite hemisphere (without SCEA implantation) were used as control for quantitative analysis. The different layers in cortex were defined according to the density of neurons using a custom Matlab script (Mathworks, USA)[27]. Analysis of the fluorescence images was done using ImageJ software (National Institutes of Health, USA). Neuronal and microglial cell densities were determined by automatic counting using ImageJ Particle Analyzer function. The parameters for counting neurons were set as follows: size = 20–200,000 μm$^2$, circularity = 0.3–1, and the threshold was set at the same level for implanted and contralateral control hemisphere. The parameters for counting microglial cells were set as follows: size = 20–200,000 μm$^2$, circularity = 0.2–1, and the threshold was set at the same level for implanted and control sides. When it came to astrocyte cells, because the GFAP labelling extended to connected networks, it was hard to discern individual cell bodies for cell counting. The intensity of astrocytes activation and accumulation was determined by calculating the proportion of the area occupied by GFAP signals above a threshold intensity level, which was the mean gray value of the pixels in layer V and VI on each image[24]. The cell density (for neurons and microglia) and signal (for astrocytes) of the implant sides was normalized to that of the control sides (average values of all animal samples at each time point). The statistical comparison between the implanted and control sides were conducted using paired $t$-tests (if the data was in a normal distribution) or related-samples Wilcoxon signed rank test (if the data was not accordance with a normal distribution), with a significance level of $p = 0.05$.

## Recording from rats

In a typical experiment, the SCEA was implanted in the left hemisphere of a rat through the minimally invasive surgery. The anesthesia was switched from isoflurane to ketamine/xylazine (100 mg/kg ketamine/10 mg/kg xylazine). Two stainless-steel screws on skull were served as reference and ground electrode respectively. A $2 \times 1.5$ mm large hole was drilled in the skull around the left-anterior corner of the SCEA, through which 1 μL of 25 mM 4-AP in saline was injected to induce seizure. Electrophysiological recordings were carried out with a 20 kHz sampling rate without notch filter using an Intan preamplifier system (RHD2164, Intan Technologies LLC., USA). Data were analyzed offline

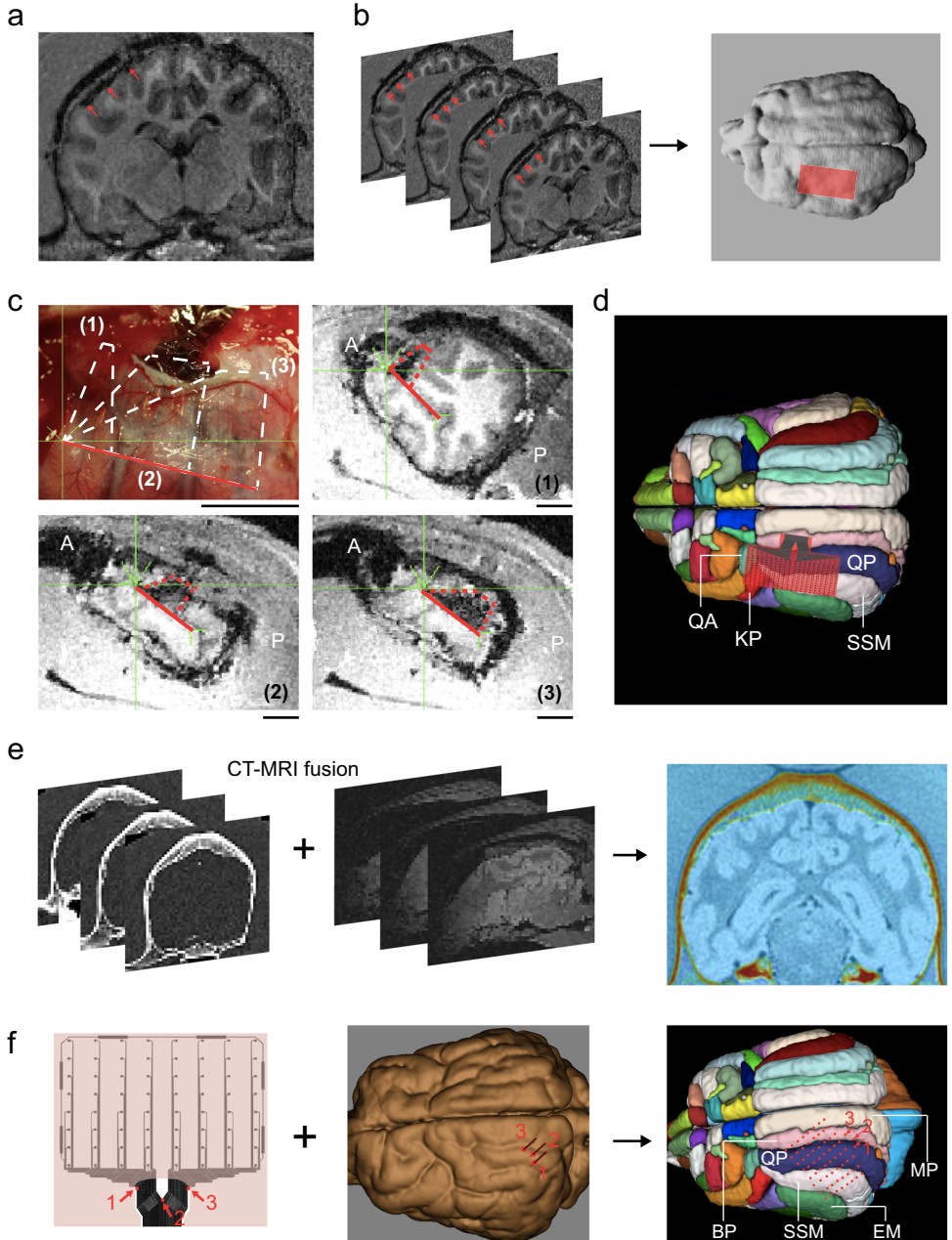

**Fig. 6 | Localization of the SCEAs. a–d** An example showing postoperative SCEA localization. **a** A postoperative T1-weighted MRI image of a beagle dog brain implanted with a 20 mm × 10 mm 256-channel Ni-SCEA. **b** 3D reconstruction of the brain and Ni-SCEA according to serial MRI images using MIPAV software (http://www.mipav.cit.nih.gov). The Ni-SCEA is schematically shown by the red rectangle. The red arrows in (**a**, **b**) point to the Ni-SCEA. **c** Three MRI images sectioned along the cortical surface of the brain from the reconstructed 3D MRI image. The portion of the Ni-SCEA sectioned in each plane (1–3) was located in the optical image acquired during surgery, where the semitransparency of the beagle dog dura enabled direct visualization of the Ni-SCEA. The solid red line indicates one edge of the Ni-SCEA, which has a length of 20 mm. 'A' indicates anterior, 'P' indicates posterior. **d** Localization of the Ni-SCEA relative to anatomical cortical structures by registering the T1 population template to the subject's T1 native space. **e, f** An example showing intraoperative localization of the SCEA using the Brainsight neuronavigation (Vet Robot) system. A 20 mm × 15 mm 64-channel SCEA was implanted subdurally into the brain of a beagle dog. **e** Overlay of CT images onto MRI images to create a 3D reconstructed image of the skull and brain. **f** Localization of the SCEA using the Brainsight™ neuronavigation (Vet Robot) system. By defining the positions of three points marked by 1–3 around the interconnection lines at the dura slit exit, the locations of the recording contacts, which are marked with red dots, relative to the anatomical structures of the brain could be determined according to the layout of the electrode design. KP coronalis posterior, QA ectolateralis anterior, QP ectolateralis posterior, SSM suprasylvian gyrus, MP marginal gyrus, BP entolateral gyrus, EM ectosylvian gyrus. The population template in panels d and f are adapted from ref. 45.

using Neuroexplorer software and custom Matlab codes (Mathworks, USA) for neural signal analysis. In Fig. 4e, the signal of channel 17 in the core area and channel 40 in the surrounding area were used for ictal PAC (11.5–58.5 s segment) and inter-ictal PAC (0-11.5 s segment) calculation. The HFO signals shown in Fig. 4g were from channel 17 from

the segment of 19 s to 19.5 s in Fig. 4b. To extract the signals of different bands, raw data were band-pass filtered for each band using a Butterworth band-pass filter in the forward and reverse directions in order to avoid phase distortion. Five independent recordings were carried out on rats with one representative example shown in Fig. 4.

## Beagle dog surgery and electrophysiological recording

Canine experiments were approved by Sincgene Institutional Animal Care and Use Committee (XNG-IAC-20210401). Due to the much thicker dura mater, SCEAs were implanted to beagles subdurally through the minimally invasive procedure in order to obtain a high-quality signal. The adult male beagles were obtained from and housed at Beijing Sincgene company, China. The general anesthesia was performed by a board-certified veterinary anesthesiologist during the whole process. Each dog was premedicated with an intramuscular injection of a mixture of dexmedetomidine (5 μg/kg) and Zoletil 50 (1 mg/kg). An indwelling needle was left in the cephalic vein of the left forearm for vascular access. Dogs were intubated after being induced to general anesthesia with propofol (1 mg/kg). The general anesthesia was maintained with inhalant isoflurane (1–2%) in oxygen (0.5 L/min) provided through an anesthesia machine (ACM 606, Beijing Aerospace Changfeng Co. Ltd, Medical Devices branch, China). During the surgery, constant rate infusion (CRI) of dexmedetomidine (1 μg/kg/h) was used for pain management, and CRI of normal saline (5 ml/kg/h) was used to maintain fluid perfusion. The animal's heart rate, respiration, oxygen saturation, tidal volume, exhalant $CO_2$, body temperature, and blood pressure were continuously monitored throughout the entire session. The animal was maintained oxygen saturation level >94% throughout the surgery. For a SCEA implantation, an incision in the midline was made and the muscles were retracted in order to expose the skull, followed by a 4 cm × 2.5 cm large craniotomy in the right hemisphere. A 6 mm long slit was carefully cut in the dura with a dural scissor, though which the SCEA strip was inserted under the dura mater onto the cortex surface. After insertion, the SCEA expanded spontaneously and fully to its original sheet shape. About 40 min later, the nitinol actuator was retracted, finishing up the minimally invasive subdural implantation of the SCEA.

The electrophysiological recordings were carried out with a stainless-steel screw in the skull near the craniotomy serving as the ground electrode, and pads located at the peripheral of the electrode array as the reference electrode. Signals were sampled at 20 kHz using an Intan preamplifier system (RHD2164, Intan Technologies LLC., USA). Signals from each channel were digitally re-referenced to the common average reference over all channels. An epoch of 15 s signal under anesthesia and emergence (6 min after discontinuing isoflurane) were used for data analysis in Fig. 5 except the coherence calculation which used an epoch of 5 s signal. Signals were analyzed using custom Matlab codes. The signal frequency domain was segmented into delta (1–4 Hz), theta (4–8 Hz), alpha (8–12 Hz), beta (12–30 Hz), gamma 1 (30–80 Hz) and gamma 2 (80–150 Hz) bands. Channel-to-channel coherence calculations were performed using the coherencyc function from chronux toolbox (http://chronux.org/)[43]. The PSD and coherence values in Fig. 5 were averaged across the frequency range for each band. The statistical significance of the coherence change between anesthesia and emergence were analyzed using Wilcoxon signed rank test owing to the fact that coherence values were not in normal distribution, with a significance level of $p = 0.05$. The statistical comparisons in Fig. 5 were performed across the frequency range for each band. Four recordings were carried out on two beagle dogs with one representative example shown in Fig. 5.

## SCEA localization

MRI and CT experiments on beagle dogs were approved by Vital Steps Institutional Animal Care and Use Committee (SC2021-11-004). MRI scans were carried out on beagles on a Siemens Prisma^fit 3.0 T MR scanner (Siemens Healthineers, Erlangen, Germany) with a four channel Tx/Rx RF coil for receiving. The MRI scans were done after the SCEA implantation for postoperative localization, and prior to SCEA implantation for intraoperative localization. Each dog was anesthetized as described above and was maintained at general anesthesia state with 1.5% isoflurane during MRI scanning. 3D reconstruction from serial MRI

images was conducted after skull stripping using the MIPAV software (http://www.mipav.cit.nih.gov). The sectioning shown in Fig. 6c was performed using the software of the Brainsight Vet Robot system (Rogure Research Inc., Canada) along anterior-posterior direction. The registration of the subject's T1 images to the atlas T1 images of population template (https://ecommons.cornell.edu/handle/1813/67018)[39] was conducted with itk-SNAP software (www.itksnap.org)[44,45].

Preoperative CT scanning of beagles, which was required for intraoperative SCEA localization, were done on Philips MX 16 Slice CT System (Philips, Netherlands). Beagles were anesthetized by intravenous injection of propofol solution (1 mg/kg) and placed into the scanner in a sphinx position. The scanning was done coronally using a standard protocol. The intraoperative localization was carried out on a Brainsight Vet Robot system (Rogure Research Inc., Canada). Prior to the surgery, CT and MRI scans of a beagle were co-registered using the Brainsight software to align the skull and the brain. During the surgery, the exposed skull was scanned by a red laser attached to the robotic arm to find the same homologous points with that on the 3D reconstructed images. This way a common 3D coordinate system was established for MRI/CT reconstruction images and the operative field of the animal. Every position on the brain could be specified correspondingly in MRI/CT images. After minimally invasive implantation of the SCEA, a needle attached to the robotic arm was used to collect the location coordinates of three points on the interconnecting lines at the dura slit exit in the common 3D coordination system. Then, the coordinates of the electrode sites were calculated according to the coordinates of the three points and the layout of the electrode design. After registering MRI images of population template onto the subject's native space, the exact locations of the electrode sites relative to the anatomical structures of the brain can be determined from the atlas on the template.

## Reporting summary

Further information on research design is available in the Nature Portfolio Reporting Summary linked to this article.

## Data availability

The source data generated in this study are provided in the Source Data. xls file. The raw data for MRI, CT and fluorescent images used for the analysis in this study are available in the Open Science Framework under ID: kucgw (https://osf.io/kucgw/). Source data are provided with this paper.

## Code availability

All the codes used for analysis have been provided on the Open Science Framework under ID: kucgw (https://osf.io/kucgw/).

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

## Acknowledgements

We thank Dr. Xiaobo Jia from School of Life Sciences at Peking University for assistance on tissue sectioning, Dr. Yuheng Lu from Institute of Automation, CAS, for help on MRI results analysis, and Peking Nanofab for the use of microfabrication instruments. This work was supported by the National Natural Science Foundation of China (Nos. T2188101, 21972005), the National Key R&D Program of China (2021YFF1200700), STI2030-Major Projects (2021ZD0202204, 2021ZD0202200), and Natural Science Foundation of Beijing Municipality (JQ20008).

## Author contributions

X.D. supervised the project. X.D. and S.W. conceived the project and designed the experiments. S.W., A.J., J.Zhu and J.Zha. fabricated and characterized the devices. S.W., A.J., X.L., X.F. and G.L. conducted the MRI and histology studies, the recordings on rats and related data analysis. S.W., A.J., H.S., P.W. and T.J. did the recordings from canine models. S.W., A.J., H.S., Z.Xu. and S.J., conducted the analysis on data from canine recordings. Y.S., Z. Xi. and A.C. did the growth of the CNT film. X.D. and S.W. wrote the manuscript with input from all authors.

## Competing interests

The authors declare no competing interests.
