## [Peer Review File · Nature Communications]

Shape-changing electrode array for minimally invasive large-scale intracranial brain activity mappingREVIEWER COMMENTS

Reviewer #1 (Remarks to the Author):

The authors demonstrate a new class of ultrathin, flexible shape-changing electrode array (SCEA) that can be inserted into the brain through a small hole in the skull or dura. After insertion the SCEA unraveled to achieve large-area electrocorticograms (ECoGs) in rat (4.6 mm x 4.6 mm, 64 electrodes) and beagle dog (20 mm x 10 mm, 256 electrodes) models. MRI studies in the rat model showed that the SCEA did not breach the blood brain barrier and caused only minor meningeal enhancement that resolved after 4 weeks and was consistent with other implantation procedures. Further, immunohistochemistry studies showed that for up to eight weeks the SCEA did not change the distribution of neurons, astrocytes and microglia, and did not disrupt the cytoarchitecture.

These SCEA allowed for detailed recordings of neural activity in both animal models. When a seizure was induced in the rat model recorded abnormalities such as spike and wave discharges and an increase in the power of low-frequency component in the power spectrum. The SCEA also demonstrated that epileptic spike amplitude was greatest at the site of injection, and that neural waves propagated from that region to distant regions. Significantly, the SCEA was able to monitor other phenomenon that are characteristic of mature brain function, including high-frequency oscillations and phase amplitude coupling. In the beagle dog model the authors observed delta, theta, alpha, beta and gamma oscillations and observed that they changed as the animal emerged from anesthesia.

This is a landmark study that could have far-reaching consequences in the areas of bioelectronics and brain-machine interfaces. The SCEA unwrapping mechanism is by itself novel, and enables brain recordings over areas that cannot be achieved by other systems. This technology allows for neuroscience studies unattainable by other means. For example, while multi-electrode arrays (MEA) have been used to measure phase amplitude coupling in neural tissue, but these studies have been confined to planar MEAs and brain organoids. The ability of the SCEA to provide real-time readouts is another benefit, and could open new doors for studying neurological diseases or implementing closed-loop bioelectronic medicines.

This study will be of great interest to the readership of Nature Communications. One small point is that the authors may wish to expand upon their Fig. 1h and comment upon their device impedance/stability over time. This small point notwithstanding, I recommend publication in strongest possible terms.

Reviewer #2 (Remarks to the Author):

This work describes the use of shape-changing electrode array, which can unfold intracranially after delivery, for ECoG brain activity mapping. This idea is interesting, but the idea was not corroborated by experimental evidence.

The biggest concern is that the described method relies on an uncontrollable unfolding process of nitinol in an intracranial space that cannot be visualized easily (e.g. optical microscopy) during delivery. Fig. 1a-c probably describes an ideal outcome that is oversimplified. According to the layout of fabrication (Fig. 1d), ECoG electrodes are only exposed to one side of the array. The unfolding process cannot control which side of the array will face the brain tissue. So by chance, 50% of times this method would yield no ECoG signals at all. The authors probably only report the best case scenario in the following figures. How many animals have the authors implanted the array and what is the percentage of successful recording?

The authors repeatedly claimed that their devices are delivered subdurally without giving a cross-section image to prove so. In contrast, the description in Methods reads the opposite: “a SCEA strip in its compressed state was slowly inserted into the epidural space”. Are the devices positioned epidurally or subdurally? How did the authors ensure that the arachnoid membrane is intact during insertion? The methods section should provide sufficient information about the craniotomy and insertion process for reproduction of the results.

The rodent cortical surface has a high curvature. Hence, to ensure that the insertion does not damage any brain tissue or blood vessels in the dura and arachnoid, the actuator must follow the curvature of the cortical surface during insertion. However, how this can be achieved in operando is not described in the manuscript. Is the actuator pre-strained prior to insertion? If so, how can the pre-strained actuator conform to the natural curvature of the cortical surface that is different for each animal and cannot be determined without removing the skull?

The necessity to remove the implanted ECoG array is another undescribed concern about this technology. The small burr hole in the cranium limits the width of the array that can be removed through it, especially considering the actuator will be removed after implantation. The array will likely have a concentrated strain near the neck at the burr hole, causing the unfolded array to be trapped inside the cranium without complete removal. Given that ECoG is usually used for intraoperative monitoring of cortical activity, the difficulty in removing the array limits the usefulness of this method.

The dotted red line in Figure 2b indicates the position of the SCEA. Is this position measured from T2-weighted MRI or simply labeled on the images? I cannot tell its position in all other images without red dot labeling, so it would be helpful for the authors to explain how they labeled it.

Fig. 3: the intensity comparison in the images does not agree with the trend in histograms. For example, GFAP signal of the implanted side is apparently higher than that in the control side in L1 in Week 2, but the histogram below shows the opposite. The GFAP signal of the implanted side is also higher than the control side in UL in Week 4, but the histogram not only shows the opposite trend but even with statistical significance. The authors need to explain this statistical significance in addition to the inconsistency with the images.

Can SCEA record ECoG chronically after implantation? Histology suggests so but there is no ECoG data showing chronic measurements.

Reviewer #3 (Remarks to the Author):

In this manuscript, Wei et al. reported on an implantation strategy that can deploy SCEA ECoG devices on the epidural/subdural space of the brain with minimal craniotomy/durotomy and therefore simplify the surgical procedures and reduce the risk of inflammatory responses and surgical lesion. The authors exploited temperature-responsive nitinol shape memory alloy as the implantation apparatus, and demonstrated efficient delivery through a small cranial opening, with the fast-response unfolding of flexible SCEA ECoG devices. The authors further showed the usefulness of the tool in detecting seizures in mice (epidural) and differentiating anesthesia/emergence states in beagle dogs (subdural). The study is meaningful and comprehensive, and I believe the work presented here can provide important insights to SCEA device deployment in both laboratory and clinical settings. I would like to recommend its publication in Nat. Commun. after minor revision. My questions/concerns are listed below:

- (1) I believe it is meaningful to quantify how effective the CNT interlayer is. It was stated in the manuscript that the stretchable CNT interlayer can maintain connectivity after mechanical deformation, but a comparison of devices with and without the CNT interlayer was not shown.
- (2) The temperature response of the shape-memory alloy is pretty fast (complete in ~10 seconds). Does it mean that the whole implantation procedure needs to be done within such a short period? It is definitely beneficial in some aspects but might pose additional challenges to the proficiency of surgeons. I wondered if it is possible to slow down the response speed of the shape recovery.
- (3) I am glad to see that the authors develop methods to locate electrodes more precisely. I wondered how precise the positioning of individual electrodes can be. Please provide more details.
- (4) A suggestion about data presentation: For the bar charts in the manuscript (Fig. 3c-f), it would be beneficial to include data points together with the error bars.

REVIEWER COMMENTS

Reviewer #1 (Remarks to the Author):

The authors demonstrate a new class of ultrathin, flexible shape-changing electrode array (SCEA) that can be inserted into the brain through a small hole in the skull or dura. After insertion the SCEA unraveled to achieve large-area electrocorticograms (ECoGs) in rat (4.6 mm x 4.6 mm, 64 electrodes) and beagle dog (20 mm x 10 mm, 256 electrodes) models. MRI studies in the rat model showed that the SCEA did not breach the blood brain barrier and caused only minor meningeal enhancement that resolved after 4 weeks and was consistent with other implantation procedures. Further, immunohistochemistry studies showed that for up to eight weeks the SCEA did not change the distribution of neurons, astrocytes and microglia, and did not disrupt the cytoarchitecture.

These SCEA allowed for detailed recordings of neural activity in both animal models. When a seizure was induced in the rat model recorded abnormalities such as spike and wave discharges and an increase in the power of low-frequency component in the power spectrum. The SCEA also demonstrated that epileptic spike amplitude was greatest at the site of injection, and that neural waves propagated from that region to distant regions. Significantly, the SCEA was able to monitor other phenomenon that are characteristic of mature brain function, including high-frequency oscillations and phase amplitude coupling. In the beagle dog model the authors observed delta, theta, alpha, beta and gamma oscillations and observed that they changed as the animal emerged from anesthesia.

This is a landmark study that could have far-reaching consequences in the areas of bioelectronics and brain-machine interfaces. The SCEA unwrapping mechanism is by itself novel, and enables brain recordings over areas that cannot be achieved by other systems. This technology allows for neuroscience studies unattainable by other means. For example, while multi-electrode arrays (MEA) have been used to measure phase amplitude coupling in neural tissue, but these studies have been confined to planar MEAs and brain organoids. The ability of the SCEA to provide real-time readouts is another benefit, and could open new doors for studying neurological diseases or implementing closed-loop bioelectronic medicines.

This study will be of great interest to the readership of Nature Communications. One small point is that the authors may wish to expand upon their Fig. 1h and comment upon their device

impedance/stability over time. This small point notwithstanding, I recommend publication in strongest possible terms.

Response: We thank the reviewer for the positive assessment on our work. In the revised manuscript, we have added a comparison between SCEAs (with CNT layer) and pure Au electrode array (without CNT layer) on the impedance change after shape transformation, to indicate the necessity of adding CNT layer in the SCEA design. We also started the impedance stability test and the data we have now indicates that the electrode array retained constant impedance values for at least two weeks (Response Figure R1). This test is continuing and we hope to report this result after getting more data and related long-term chronic *in vivo* recordings somewhere else in the future.

New Supplementary Fig. S3. Comparison between SCEAs and pure Au electrode array on impedance change after one shape transformation cycle (sheet to strip, and then back to sheet).

Response Figure R1. *In vitro* impedance stability of SCEAs measured in PBS. n=88 from 6 SCEAs.

Reviewer #2 (Remarks to the Author):

This work describes the use of shape-changing electrode array, which can unfold intracranially after delivery, for ECoG brain activity mapping. This idea is interesting, but the idea was not corroborated by experimental evidence.

The biggest concern is that the described method relies on an uncontrollable unfolding process of nitinol in an intracranial space that cannot be visualized easily (e.g. optical microscopy) during delivery. Fig. 1a-c probably describes an ideal outcome that is oversimplified. According to the layout of fabrication (Fig. 1d), ECoG electrodes are only exposed to one side of the array. The unfolding process cannot control which side of the array will face the brain tissue. So by chance, 50% of times this method would yield no ECoG signals at all. The authors probably only report the best case scenario in the following figures. How many animals have the authors implanted the array and what is the percentage of successful recording?

Response: We thank the reviewer for raising this concern, and apologize for any lack of clarity in the original manuscript. The nitinol shape actuator was attached to the backside of the electrode array and remained in contact with the backside throughout the entire compression and expansion process. After compression, the electrode/nitinol complex formed a strip. By appropriately orienting the strip during the implantation, we can make sure that the active site side faces the brain tissue. We have added these details in legend of Supplement Fig. S1c and the Methods section (Page 16) for clarification.

Supplementary Figure S1c. Schematics showing the bonding and compression of the CNT/Au array and nitinol complex. The nitinol shape actuator was attached to the backside of the CNT/Au array using polyethylene oxide (PEO) solution, and remained on the backside of the array throughout the deformation process, as depicted in the figure.”

Page 16: “Since the nitinol actuator was attached to and remained on the backside of the electrode array, by appropriately orienting the strip during the implantation, it can be ensured that the active site side faces the brain tissue.”

In addition, it is important to note that the nitinol shape actuator only expands from the middle to the periphery during the deformation process, with no rotation or turning during the shape changing process. This can be seen from the recorded movie showing the shape changing process *in vitro* (Supplementary Video). We have also added two new Supplementary Figures which include pictures of the devices during the expansion both *in vitro* (Supplementary Figure S2) and in subdural space of a beagle brain (Supplementary Figure S7), in order to better illustrate the expansion process. This way by controlling the orientation of the electrode during insertion, we were able to ensure that the electrode always faced the brain tissue, allowing us to achieve consistent recordings during each implantation. In fact, we have implanted 9 devices into rats and 4 devices into beagles, and all of them successfully recorded ECoG signals.

Once again, we thank the reviewer for their valuable feedback, which has helped us to clarify this important aspect of our work.

New Supplementary Figure S2. Deployment of a SCEA *in vitro*. The SCEA was placed on the surface of an agar gel surrounded by 37°C saline. Scale bar, 3 mm.

New Supplementary Figure S7. Deployment of a SCEA subdurally in a beagle brain. The pictures were taken at different time points during the deployment. Scale bar, 5 mm.

The authors repeatedly claimed that their devices are delivered subdurally without giving a cross-section image to prove so. In contrast, the description in Methods reads the opposite: “a SCEA strip in its compressed state was slowly inserted into the epidural space”. Are the devices positioned epidurally or subdurally? How did the authors ensure that the arachnoid membrane is intact during insertion? The methods section should provide sufficient information about the craniotomy and insertion process for reproduction of the results.

Response: We thank the reviewer for raising this question. We have achieved both epidural and subdural implantation for SCEAs. Specifically, in our rodent surgeries, we implanted the SCEAs into the epidural space, while for beagle dogs, the SCEAs were implanted into the subdural space. The reason for this difference is that the dura mater in rodents is thin and soft, and does not significantly attenuate the signal. However, in larger animals, the dura mater is much thicker (Response Figure R1, $233 \pm 71 \mu\text{m}$ for dogs *versus* $49 \pm 15 \mu\text{m}$ for rodents¹) and will lead to significant attenuation of the ECoG signals. For large animal models, subdural implantation gives signals with better quality, higher resolution and wider bandwidth.

The successful subdural implantation can be seen in the optical images in Fig. 5b, where a SCEA was implanted into the subdural space of a beagle dog, and the images were taken before and after withdrawal of the nitinol shape actuator. The semi-transparent dura mater can be seen on top of the nitinol shape actuator and the electrode array. As suggested by the reviewer, we have added more details regarding the craniotomy and insertion process in the Methods section (Page 16, 20), and also revised the main text to clarify the implantation type (epidural or subdural) (Page 10).

Page 10 “The strategy of using the SCEAs for minimally invasive intracranial brain activity mapping can also be applied subdurally in large animal models. The larger thickness of the dura mater in large animals causes more severe neural signal attenuation than in rodents. Hence, compared to epidural recordings, subdural recordings are more desirable for large animal models.”

Page 16 “SCEAs were implanted to rats epidurally through the minimally invasive procedure.”

Page 20 “Due to the much thicker dura mater, SCEAs were implanted to beagles subdurally through the minimally invasive procedure in order to obtain a high-quality signal.”

The arachnoid mater is located between the dura mater and pia mater, which can remain undamaged for epidural implantation. Subdural implantation can cause temporary damage to the arachnoid layer, which is a common challenge for all subdural recording methods.

The arachnoid mater comprises two layers: the outer arachnoid barrier cell layer and the inner arachnoid trabeculae layer (Response Figure R1)². The arachnoid barrier cell layer is located beneath the dura mater and has a stronger adhesion to the dura than to the rest of the arachnoid, making it hard to separate it from the dura. The arachnoid trabeculae layer consists of web-like chordae that traverse the subarachnoid space and has a looser arrangement than the arachnoid barrier cell layer³. During subdural implantation, SCEAs were inserted into the subarachnoid space, which is under the outer layer of the arachnoid mater. The arachnoid trabeculae layer may incur some extent of damage, which is a common problem encountered by all subdural electrodes. However, different from other subdural electrode array, the SCEAs had a mesh structure. The holes in the mesh structures allow for cerebrospinal fluid circulation and mass exchange in the subarachnoid space, which is advantageous for the regeneration of the arachnoid trabeculae layer.

Table 1. Histologic characteristics of the dura according to species

	No. of samples	Dural thickness (μm)		No. of layers	Fibroblast orientation		
		mean	1 SD		no distinguishable layers	periosteal layer	meningeal layer
Human	3	564	50	3	—	slightly haphazard	slightly haphazard
Cow	4	311	87	3	—	longitudinal	longitudinal
Cat	3	201	78	3	—	slightly haphazard	longitudinal
Dog	4	233	71	3	—	slightly haphazard	longitudinal
Goat	3	284	57	2	longitudinal	—	—
Horse	3	313	64	2	slightly haphazard	—	—
Pig	4	304	77	3	—	slightly haphazard	longitudinal
Rabbit	3	99	32	3	—	slightly haphazard	longitudinal
Rat	4	49	15	2	slightly haphazard	—	—
Sheep	3	234	91	2	slightly haphazard	—	—

Response Figure R1. (Top) The histologic characteristics of the dura mater of various species showing the dura mater thickness¹; (Bottom) schematic showing the structure of the brain surface².

The rodent cortical surface has a high curvature. Hence, to ensure that the insertion does not damage any brain tissue or blood vessels in the dura and arachnoid, the actuator must follow the curvature of the cortical surface during insertion. However, how this can be achieved in operando is not described in the manuscript. Is the actuator pre-strained prior to insertion? If so, how can the pre-strained actuator conform to the natural curvature of the cortical surface that is different for each animal and cannot be determined without removing the skull?

Response: We thank the reviewer for raising this question. We didn't pre-strain the actuators in our work. The deployment of the SCEAs occurred in the space between the skull and dura mater (for epidural implantation) or between the dura mater and cortical surface (for subdural

implantation). As described in the Methods section, the animals were pretreated with mannitol before SCEAs injection. Mannitol is known to be an osmotic agent that has the ability to reduce the water content and volume of the brain. Studies have reported that mannitol can lead to a reduction of approximately 0.8% in the size of the human brain⁴. This reduction in brain volume creates sufficient space for the deployment of SCEAs without causing significant compression or displacement of brain tissue. In addition, with a diameter of 100 and 250 μm , the nitinol shape actuator is easily bendable. Under the constriction of the skull or dura mater, the SCEAs can bend slightly to follow the small curvature of the brain (Response Figure R2). After completion of the deployment, the electrode array was released from the nitinol shape actuator, and due to its ultrasmall thickness (4 μm) and mesh structure, the electrode array would conformally adhered to the surface of the brain, as shown by the picture in Fig. 1j and 5b. That is to say, the conformal interfacing between the electrode array and the brain surface was actually formed under the surface tension due to the ultrathin nature and mesh structure of the electrode array film.

We have added more details regarding this point on Page 6 in the revised manuscript.

Page 6 “After washing with saline for ~40 mins, the SCEAs were released from the shape actuator which was then slowly retracted through the small cranial opening. To confirm the placement of the SCEA, we performed a craniotomy after the SCEA was implanted. The SCEA transformed its shape from a strip to its fully deployed state and the 4 μm thickness of the electrode array resulted in a conformal interface with the epidural cortical surface...”

Response Figure R2. Schematics showing the deployment process.

The necessity to remove the implanted ECoG array is another undescribed concern about this technology. The small burr hole in the cranium limits the width of the array that can be removed

through it, especially considering the actuator will be removed after implantation. The array will likely have a concentrated strain near the neck at the burr hole, causing the unfolded array to be trapped inside the cranium without complete removal. Given that ECoG is usually used for intraoperative monitoring of cortical activity, the difficulty in removing the array limits the usefulness of this method.

Response: We thank the reviewer for bringing up this issue. We acknowledge that removing the SCEAs from the surface of the brain without any assistance could be challenging. This is mainly because the electrode array is conformably and tightly adhered to the brain surface, which requires large force to separate the film, thus may cause the breakage of the electrode array, considering its small thickness and the mesh structure. A possible method is to make an opening on the skull at the other end of the electrode array, through which the electrode array can be peeled off from the brain. Operating from the two openings corresponding to the two ends, the electrode array could be safely separated from the brain and dragged out through a burr hole. The electrode array is highly flexible so it should be able to be squeezed and pass through the burr hole.

We would like to point out that for acute intraoperative monitoring, it may not be necessary to remove the nitinol actuator. In our original design, we connected the nitinol actuator to the electrode using a water-soluble polymer PEO so that the actuator and electrode could be separated after deformation. As an alternative, we could use a non-soluble polymer to connect the actuator and electrode. This way the electrode array can remain bonded to the nitinol actuator. After finishing the recording, the electrode array can be retracted together with the nitinol actuator, in a way similar to how a stent is removed from a blood vessel.

Our SCEAs strategy enables the implantation of ultra-flexible, even mesh-structured electrode arrays onto a large-area cortex surface. The shape actuator can be used with various electrode and substrate materials, increasing its compatibility and versatility. Another solution to solve the removal issue is to combine the idea of transient electronics in the SCEA design. By using biodegradable conductive and insulation materials, the implanted electrode array can resolve under physiological conditions, thus avoiding the necessity to remove the implants. However, it is important to note that much effort is still required to ensure that biodegradable electrodes are compatible with the large and extreme deformations that occur during the fabrication and surgery processes. Exploring the compatibility of biodegradable electrodes with the large and extreme

deformations that occur during the fabrication and surgery processes is one of promising future research directions. Such investigations hold great potential for further improving the performance of SCEAs and expanding their applications in both fundamental and translational neuroscience.

We have added the discussion on the removal issue on Page 14 of the revised manuscript.

Page 14 “. It is noted that the conformal and tight brain interface and the ultrathin nature of the SCEAs pose challenges on minimally invasive removal of the implants through the small opening in the skull or dura. For future development, combining the transient electronics idea with the SCEA strategy could provide a promising solution for safe removal of the implants without the need for a second surgery.”

The dotted red line in Figure 2b indicates the position of the SCEA. Is this position measured from T2-weighted MRI or simply labeled on the images? I cannot tell its position in all other images without red dot labeling, so it would be helpful for the authors to explain how they labeled it.

Response: We thank the reviewer for bringing up this question. We appreciate the opportunity to clarify our methods. The SCEAs were invisible on MRI scans and no obvious damage was induced for the brain underneath the SCEAs. So it is hard to tell the position of the SCEAs from MRI pictures. The dotted red lines indicating the position of the SCEAs were determined according to the implantation position and the geometry of the mesh SCEA electrode. We have added this clarification in the legend of Fig. 2 in the revised manuscript.

Page 31 “The dotted red line shows the position of the SCEA which was determined from the implantation position and the geometry of the array.”

As described in the Methods section, the size of the SCEA is 6 mm by 6 mm (Figure S4). For each implantation, a small skull opening measuring 2 mm × 0.8 mm was made on the right skull, with the lower left corner of the opening located 1 mm anterior and 3 mm lateral from the lambda (Methods). The central location of each electrode implantation was approximately 4 mm apart from the middle line of the rat brain. When labeling the red dotted line in the picture, we selected the central location of each electrode as a reference point and then expanded by ~3 mm arc length in the medial and lateral directions.

Fig. 3: the intensity comparison in the images does not agree with the trend in histograms. For example, GFAP signal of the implanted side is apparently higher than that in the control side in L1 in Week 2, but the histogram below shows the opposite. The GFAP signal of the implanted side is also higher than the control side in UL in Week 4, but the histogram not only shows the opposite trend but even with statistical significance. The authors need to explain this statistical significance in addition to the inconsistency with the images.

Response: We thank the reviewer for raising this important question. We looked into the data carefully and confirm that for the specific rat shown in the original Fig. 3b, the GFAP signal in L1 at week 2 on the implanted side was indeed higher than that on the control side. The ratio of the implanted side to the control side is 1.33. However, it is important to note that this value is within the variations among different subjects (see Fig. 3d). The statistical analysis between these two groups showed no significant difference (Fig. 3d). We thank the reviewer for pointing out this issue, and to avoid any confusion, we have replaced the images with data from another rat.

Regarding the GFAP signal in UL during week 4, for the specific subject shown in Fig. 3b, the normalized signals for the implanted and control parts are 0.70 and 1.01, respectively. The ratio between the two signals is 0.69, which actually aligns with the histogram data. The visual differences observed by the reviewer could be due to the difference in background signal intensity. we would like to take this opportunity to explain our analysis method in detail. The regions of interest in the right and left, corresponding to the implanted and control side respectively, were far from each other. We were not able to obtain both images at one scan. Instead, the two images of the implanted and controlled side were collected at two different scans. As a result, the background fluorescence intensity was different among the two image and we could not directly compare the absolute fluorescence intensity of the GFAP signals between the implanted and control hemisphere. In addition, because the GFAP labeling extended to connected networks, it was difficult to discern individual cell bodies for cell counting.

To address these issues, we used the analysis methods published previously (Reference 24 in manuscript). Specifically, the intensity of astrocytes activation and accumulation was determined by calculating the proportion of the area occupied by GFAP signals above a threshold intensity level, which was the mean gray value of all the pixels in layer I, II, III and IV on each image. The pixels with intensity above the mean were defined as signals. It is important to note that for each

image, there could be a different threshold value. This may account for the inconsistency observed by the reviewer.

For better data presentation, we overlaid all the individual data points on the box plots in Fig 3c-h. The methods were also revised to provide more details in the revised manuscript. (Page 19)

New Fig. 3 c-h | Individual data points were overlaid on c-h to provide a more detailed data presentation.

Page 19 “When it came to astrocyte cells, because the GFAP labeling extended to connected networks, it was hard to discern individual cell bodies for cell counting. The intensity of astrocytes activation and accumulation was determined by calculating the proportion of the area occupied by GFAP signals above a threshold intensity level, which was the mean gray value of all the pixels in layer I, II, III and IV on each image²⁴. It is noted that for each image, there could be a different threshold value.”

Can SCEA record ECoG chronically after implantation? Histology suggests so but there is no ECoG data showing chronic measurements.

Response: Thank you for raising this important question. We agree that chronic stability of ECoG recordings is crucial for future BMI applications, and we have initiated the chronic *in vivo*

recording tests. However, it is important to note that conducting such tests requires substantial effort and time, particularly when ensuring an adequate number of samples for reliable results. Furthermore, we are actively working on addressing additional aspects, such as the secure fixation of electrode headsets on the skull and implementing suitable protective measures for the electrode headsets in large animal models. These endeavors are essential for our system's successful integration and long-term functionality.

As mentioned earlier by the reviewer, that ECoG is conventionally employed for intraoperative monitoring of cortical activity in epilepsy foci localization, which does not necessitate chronic implantation. We propose that our SCEAs strategy offers a viable solution to mitigate the operative risks inherent in this procedure. By utilizing SCEAs, we aim to minimize invasiveness and potential complications associated with traditional ECoG electrode arrays, thereby enhancing the safety undergoing these procedures.

In the future, once we have successfully validated the chronic neural recording capability of our device (which is beyond the scope of this manuscript), we envision that it could have broad applications in fundamental brain research and BMIs. We appreciate the reviewer's interest in our work and remain committed to further testing and validation.

Reviewer #3 (Remarks to the Author):

In this manuscript, Wei et al. reported on an implantation strategy that can deploy SCEA ECoG devices on the epidural/subdural space of the brain with minimal craniotomy/durotomy and therefore simplify the surgical procedures and reduce the risk of inflammatory responses and surgical lesion. The authors exploited temperature-responsive nitinol shape memory alloy as the implantation apparatus, and demonstrated efficient delivery through a small cranial opening, with the fast-response unfolding of flexible SCEA ECoG devices. The authors further showed the usefulness of the tool in detecting seizures in mice (epidural) and differentiating anesthesia/emergence states in beagle dogs (subdural). The study is meaningful and comprehensive, and I believe the work presented here can provide important insights to SCEA device deployment in both laboratory and clinical settings. I would like to recommend its publication in Nat. Commun. after minor revision. My questions/concerns are listed below: (1) I believe it is meaningful to quantify how effective the CNT interlayer is. It was stated in the manuscript that the stretchable CNT interlayer can maintain connectivity after mechanical deformation, but a comparison of devices with and without the CNT interlayer was not shown.

Response: We thank the reviewer for the positive assessment on our work. The comparison between SCEAs (with CNT interlayer) and pure Au electrode (without CNT interlayer) was added as new Supplementary Figure S3. Related results were also added in the main text (Page 6) of the revised manuscript.

New Supplementary Figure S3. Comparison between SCEAs and pure Au electrode array on impedance change after one shape transformation cycle (sheet to strip, and then back to sheet).

Page 6 “After the SCEAs transformed shape to a strip and back to a sheet, due to the extreme strain and deformation experienced by the SCEAs, about 7.89% (n= 241 from 8 devices) of the channels with 1-kHz impedance below 1 M Ω increased impedance beyond 1 M Ω . While for pure Au electrode array without CNT layer, a much larger fraction of channels (40.28 %, n=72 from 5 devices) increased impedance beyond 1 M Ω (Fig. 1g, h, Supplementary Fig. S3), indicating the necessity of adding CNT layer in the SCEA design.”

(2) The temperature response of the shape-memory alloy is pretty fast (complete in ~10 seconds). Does it mean that the whole implantation procedure needs to be done within such a short period? It is definitely beneficial in some aspects but might pose additional challenges to the proficiency of surgeons. I wondered if it is possible to slow down the response speed of the shape recovery.

Response: We thank the review for bringing up this important question. The shape changing process occurs rapidly and typically completes within approximately 10 seconds. To provide a sufficient time window for the insertion *in vivo*, we applied additional water-soluble polymer PEO on the nitinol/array complex strip. The presence of PEO introduces a delay in the start of deployment and decelerates the deployment process due to the time required for PEO dissolution. Through this approach, the PEO effectively postpones the initiation of deployment for several minutes. Consequently, a sufficient time window is created, allowing for the insertion of SCEAs *in vivo*.

To further fine-tune the timing of the deployment process *in vivo*, we can explore the modification of various properties of the water-soluble polymers, including their concentration, amount, and solubility. By altering these parameters, the time window for insertion can be adjusted, providing better control over the deployment and positioning of the electrodes during implantation. We have elaborated on this point on Page 5, 6 in the revised manuscript.

Page 5, 6 “To provide a sufficient time window for SCEAs insertion *in vivo*, additional water-soluble PEO was applied on the nitinol/array strip. The presence of PEO introduced a delay in the

start of deployment and decelerated the deployment process due to the time required for PEO dissolution. Through this approach, the deployment was effectively postponed and slowed down for several minutes (Supplementary Fig. S2, Supplementary Video).”

(3) I am glad to see that the authors develop methods to locate electrodes more precisely. I wondered how precise the positioning of individual electrodes can be. Please provide more details_.

Response: We thank the reviewer for raising this important question. We conducted a new localization test with procedures same as described in our manuscript on a beagle dog's skull model to address the query. As described in the manuscript, with the Brainsight Vet Robot system, a common 3D coordinate system was established for MRI/CT reconstruction images and the operative field of the animal. The interconnecting lines at the dura slit exit are visible. By defining the positions of three points on the interconnecting lines (for example, point 1, 2 and 3 in Response Figure R1a) at the dura slit exit in the common 3D coordination system, the coordinates of the active sites were calculated according to the coordinates of the three points and the layout of the electrode design. After registering MRI images of population template onto the subject's native space, the exact locations of the electrode sites relative to the anatomical structures of the brain can be determined from the atlas on the template.

For the test on the skull, because all the active sites were visible, we can compare the collected coordinates with the values determined from the procedure described above (defined as calculated values). The calculated coordinates (red dots) and the collected coordinates (blue dots) are presented in Response Fig. R1c. The resulting error ranges from approximately 8.80% to 14.67%.

Response Figure R1. Accuracy test for the intraoperative localization method.

(4) A suggestion about data presentation: For the bar charts in the manuscript (Fig. 3c-f), it would be beneficial to include data points together with the error bars.

Response: We thank the reviewer for the valuable suggestion. In the revised manuscript, we have overlaid all the data points on the box plots in Fig 3c-h, in order to provide a detailed data presentation.

New Fig. 3 c-h | Individual data points were overlaid on c-h to provide a more detailed data presentation.

Reference:

1. Kinaci, Ahmet, et al. "Histologic comparison of the dura mater among species." *Comparative medicine* 70.2 (2020): 170-175.
2. Abrey, L.E., Chamberlain, M. and Engelhard, H. eds., "Leptomeningeal metastases." Springer Science & Business Media (2005): 4-5.
3. Lu, S., A. Brusica, and F. Gaillard. "Arachnoid membranes: crawling back into radiologic consciousness." *American Journal of Neuroradiology* 43.2 (2022): 167-175.
4. Videen, Tom O., et al. "Mannitol bolus preferentially shrinks non-infarcted brain in patients with ischemic stroke." *Neurology* 57.11 (2001): 2120-2122.

REVIEWER COMMENTS

Reviewer #1 (Remarks to the Author):

The authors have addressed the reviewers' comments. I recommend acceptance with no need for further revision.

Reviewer #2 (Remarks to the Author):

Please see the attachment.

While I appreciate the detailed response by the authors, I maintain serious concerns about the overall scientific rigor, transparency, and methodological validity of this revised manuscript, rendering it unsuitable for publication in a prestigious journal such as *Nature Communications*. The process of nitinol unfolding in an intracranial space, a central element of this study, lacks clear and direct *in vivo* visualization under the cranium, raising doubts about its control and reproducibility. Moreover, the authors' reliance on *in vitro* demonstrations, post-craniotomy imaging, and indirect assumptions about the SCEA location in MRI images further erodes the manuscript's credibility. Another aspect that warrants strong criticism is the proposed epidural versus subdural SCEA delivery, where the authors' arguments have been found to contain inaccuracies and fail to address potential damage caused by the SCEA operation. The justification for how SCEA conforms to the brain surface curvature by invoking capillary forces appears flawed and scientifically questionable, particularly considering the presence of cerebrospinal fluid. Additionally, there is also a disturbing lack of precision in identifying SCEA positioning in MRI scans, an issue that resonates with my critique about nitinol unfolding. Lastly, the authors' method for analyzing histology data is questionable, both in terms of visual interpretation and quantification of astrocyte activation intensity. The use of mean gray value as a threshold could potentially suppress actual signals, leading to a misrepresentation of astrocyte activation levels. Taken together, these issues seriously undermine the scientific integrity of the manuscript and therefore, it cannot be endorsed for publication in its current form. My detailed, point-by-point responses to the authors' rebuttal are provided below.

1. My primary critique of this research paper, as I have formerly expressed, pertains to the **uncontrollable unfolding process of nitinol in an intracranial space** that cannot be visualized easily (e.g. optical microscopy) during delivery. The authors have yet to adequately address this critique in their response letter, as will be comprehensively elucidated in the following text.

In response to this critique, the authors have provided supplementary materials - namely Figure S2 and an accompanying video - in an effort to elucidate the unfolding process. This initiative is indeed appreciated. Nonetheless, it is imperative to underscore that these materials were obtained under *in vitro* conditions, as admitted by the authors. Consequently, it remains an open question as to whether this unfolding process, while meticulously visualized *in vitro*, would proceed identically within the intracranial space *in vivo*.

The authors have also supplemented the manuscript with Figure S7, which depicts the deployment of a SCEA within a beagle's brain. Yet, the description provided for this figure leaves a considerable degree of ambiguity regarding whether this deployment was executed within the intact intracranial space, or post-craniotomy. Inferred from the caption of Fig. 5b ("The craniotomy caused some bleeding, which left some blood on the surface of the dura mater"), the latter seems to be the case, which, if true, would mean the images captured post-craniotomy are of limited utility in validating the nitinol unfolding process within an intact intracranial space. The confines of the intracranial space beneath an intact cranium inherently impose limitations on the unfolding of the SCEA post-delivery. Conversely, these limitations are evidently absent when a craniotomy is performed.

In my initial review, I had assumed that the MRI images in Figure 2 were indicative of the SCEA's location. However, the authors' response indicates that the depiction of the SCEA's location was inferred rather than directly imaged - an approach that introduces subjectivity and detracts from the scientific rigor expected in such illustrations.

In summary, the authors' attempts to address my primary critique have not succeeded in presenting conclusive evidence validating the controllability of the nitinol unfolding process within an intracranial space. The data provided either fall under *in vitro* studies or *in vivo* studies conducted post-craniotomy, both of which do not faithfully replicate the true conditions under a mostly intact skull aiming for minimal invasiveness. In light of the limitations posed by MRI in providing direct imaging of SCEA, owing to the presence of metals, I would recommend the authors consider leveraging high-resolution uCT techniques for *in situ* visualization of the unfolding process. It is critical that such data be included to comprehensively substantiate the unfolding process, which forms the backbone of this research paper and influences all subsequent data and claims.

2. In the authors' reply concerning the matter of epidural versus subdural delivery of the SCEA, **they have resorted to some factual inaccuracies to validate their implantation methodology.** To be specific, the authors argue that "*Subdural implantation can cause temporary damage to the arachnoid layer, which is a common challenge for all subdural recording methods.*" However, this assertion is fundamentally inaccurate.

Typically, ECoG arrays, especially in human applications, are directly placed on the leptomeninges surface, where the leptomeninges encompass both the arachnoid and pia mater. Several publications, including *Biomedical Microdevices*, 13, 59–68, 2011, have demonstrated that conventional ECoG arrays do not inflict acute damage to the leptomeninges, although long-term deployment of ECoG arrays may induce leptomeninges thickening. Another study (*Sensors* 2021, 21, 178) indicated that the arachnoid membrane exhibits local depression due to the physical presence of grid electrodes while in the subdural space, but the arachnoid membrane is not ruptured ("While in the subdural space, tissue sections showed that arachnoid membrane was locally depressed by the occupying effect of grid electrodes.").

Should the SCEA insertion and unfolding consistently inflict damage on the arachnoid layer, it points towards two unignorable facts: 1) The SCEA device, when in operation, incurs greater harm than a conventional ECoG device, and 2) The SCEA device's unfolding process within the subdural space lacks sufficient control (thus confirming my previous comment), resulting in the rupture of the leptomeninges and blood vessels.

The invasive impact of ECoG is exemplified in the authors' own data of Fig. 5b, where substantial bleeding is observed post nitinol retraction. For better clarity, I have highlighted these newly developed hemorrhagic spots with white arrows below. Given the gravity of these potential damage to the brain, they warrant immediate remediation before the publication of this method.

3. In their responses to my comments regarding how the SCEA conforms to the cortical surface curvature, the authors referenced some facts that appear to be either irrelevant or inaccurately represented.

They state that "Studies have reported that mannitol can lead to a reduction of approximately 0.8% in the size of the human brain⁴. This reduction in brain volume creates sufficient space for the deployment of SCEAs without causing significant compression or displacement of brain tissue." Upon reviewing the referenced source (reference 4), it became clear that the 0.8% reduction pertains to the overall brain volume. From this data, one can conduct a basic mathematical analysis to determine the magnitude of size reduction across each dimension:

$$1-(1-0.8\%)^{1/3} = 0.27\%$$

Assuming a dorsoventral dimension of 1 cm for a mouse brain, this shrinkage corresponds to the creation of an additional intracranial space amounting to 1 cm x 0.27% = **27 μ m**. Even when contrasted with the diameter of nitinol, which is 100 μ m, this added space is relatively insignificant. Hence, **it is not scientifically sound to use this minuscule reduction in brain volume as a justification for facilitating the creation of space necessary for intracranial unfolding.**

In addition to the previously mentioned contention, the authors present another argument, invoking the principle of capillary forces, to elucidate how the SCEA can adhere conformally to the brain surface: "due to its ultrasmall thickness (4 μ m) and mesh structure, the electrode array would conformally adhered to the surface of the brain, as shown by the picture in Fig. 1j and 5b. That is to say, the conformal interfacing between the electrode array and the brain surface was actually formed under the surface tension due to the ultrathin nature and mesh structure of the electrode array film."

This assertion, too, is scientifically flawed. Surface tension typically arises at the air-liquid interface, where an imbalance between adhesive and cohesive forces results in a net force known as surface tension, as detailed in the book "Intermolecular and Surface Forces" by J. N. Israelachvili. The intracranial space is filled with cerebrospinal fluid (CSF). Under the assumption that only a minor incision is made in the cranium to allow for SCEA insertion, it is reasonable to predict that this small opening would not lead to a significant drainage of the CSF (if it does, this would be considered as a major drawback of this method and should thus be explicitly mentioned).

Consequently, **the deployed SCEA within the intracortical space would be exposed to CSF on both sides, counterbalancing the surface tension.** A straightforward analogy can illustrate this concept: while cling wrap can adhere to the surface of food items owing to capillary forces when in air, this adhesion does not occur when the entire system is submerged in water.

4. The authors' arguments about removing the SCEA from the surface of the brain do not entirely address the complexities inherent to this issue.

First, the comparison between the removal of SCEA and the removal of a stent from a blood vessel does not take into account the vast differences in scale and complexity between these procedures. A stent, when placed in a blood vessel, encounters a relatively uniform and predictable structure. The blood vessel is typically cylindrical with a relatively consistent diameter, which significantly simplifies the process of stent removal. Conversely, the SCEA is implanted within the irregular, convoluted topography of the brain, with its myriad folds and crevices. Moreover, the brain's tissue is markedly more sensitive and prone to damage compared to the walls of a blood vessel. The sheer thinness and delicate mesh structure of the SCEA, as acknowledged by the authors, make it extremely vulnerable to breakage during removal. This could potentially result in fragments of the array being left behind in the brain, which poses serious risks and requires larger-area craniotomy for retrieval. Furthermore, most stents are designed for permanent implantation, which eliminates the issue of removal altogether.

In addition, in light of this response and another response later to chronic studies, the authors seem to suggest the SCEA technology is primarily intended for single-time-point, intraoperative monitoring. If this is indeed the case, then it becomes imperative that the SCEA be removed post-operation to prevent any possible chronic damage - an aspect that has not been sufficiently addressed in the study. If the argument is that the subject animal would be sacrificed after the experiment, thereby negating the need for SCEA removal, it calls into question the necessity of minimally invasive implantation. If it is a terminal experiment, craniotomy could be performed, as this invasive procedure would not factor into the long-term welfare of the subject.

5. The response by the authors regarding the positioning of the SCEA in MRI images raises some concerns about the precision and objectivity of this method. By stating that the SCEAs are "invisible" in the MRI scans and that their position was determined based on the "implantation position and the geometry of the array," it appears that the positioning of the SCEA is marked based on **an assumption rather than precise, empirical, and objective data.**

This method of manually labeling the position of the SCEA seems to rely heavily on subjective judgement and is therefore prone to inconsistencies or errors, especially when different individuals are involved in the process. It also suggests a lack of rigor, as subjective human estimations are inherently less reliable and less precise than measurements made with objective, scientifically verifiable methods. This lack of rigor is consistent with my first critique above, highlighting the **uncontrollable unfolding process of nitinol in an intracranial space** that is not guided by intraoperative imaging but only relies on *ex vivo* experiments or *in vivo* experiments post-craniotomy.

The authors' reliance on an approximation of the SCEA's position in the MRI images does NOT align with best scientific practices, which emphasize precision, replicability, and objective measurements. Given the potential implications of inaccuracies in the SCEA positioning, it is critical that a more objective and quantifiable method of determining the SCEA position is adopted. A more reliable approach could involve using a contrast agent or other MRI-visible markers incorporated into the SCEA design that could be accurately detected in the MRI scans. Alternatively, as suggested above, the authors should consider leveraging high-resolution uCT techniques for *in situ* visualization of the metal components in SCEA under the intact cranium.

6. Regarding the analysis method used for histology data, I have attached the Week-4 GFAP data below and have the editor and other reviewers (maybe additional reviewers too) for everyone's judgment. The authors assert that the normalized signals for the implanted and control sections are 0.70 and 1.01, respectively, within the UL layer, thus indicating a higher GFAP signal in the control as compared to the implanted section. Contrary to the authors' interpretation, the image below clearly illustrates a more pronounced GFAP signal (in green) in the implanted section than in the control part.

I would also like to question the authors' analysis method for quantifying the intensity of astrocyte activation and accumulation. The authors state that "the intensity of astrocytes activation and accumulation was determined by calculating the proportion of the area occupied by GFAP signals above a threshold intensity level, which was the mean gray value of all the pixels in layer I, II, III and IV on each image. The pixels with intensity above the mean were defined as signals." Inherent to this methodology is **a systematic issue that may inadvertently skew the results**. Specifically, the use of a threshold intensity level based on the mean gray value might inadvertently suppress the actual signal, especially in images where GFAP signals are highly represented. Consequently, images with higher GFAP signal intensity would naturally yield a higher mean gray value, thereby elevating the threshold. In turn, this could result in fewer pixels being defined as signals, potentially underestimating the true intensity of astrocyte activation and accumulation. This approach may thus obscure the actual extent of astrocyte activation and accumulation, leading to

potentially misleading conclusions. I believe this critical aspect warrants a more in-depth discussion and perhaps, a revision in the methodology.

Reviewer #3 (Remarks to the Author):

The authors very well addressed the question raised and I am satisfied with the current version of the manuscript. I would like to recommend its publication in Nat Commun as is.

Response to Reviewer #2 Comments:

Response to general comments:

While I appreciate the detailed response by the authors, I maintain serious concerns about the overall scientific rigor, transparency, and methodological validity of this revised manuscript, rendering it unsuitable for publication in a prestigious journal such as *Nature Communications*. The process of nitinol unfolding in an intracranial space, a central element of this study, lacks clear and direct in vivo visualization under the cranium, raising doubts about its control and reproducibility. Moreover, the authors' reliance on in vitro demonstrations, post-craniotomy imaging, and indirect assumptions about the SCEA location in MRI images further erodes the manuscript's credibility. Another aspect that warrants strong criticism is the proposed epidural versus subdural SCEA delivery, where the authors' arguments have been found to contain inaccuracies and fail to address potential damage caused by the SCEA operation. The justification for how SCEA conforms to the brain surface curvature by invoking capillary forces appears flawed and scientifically questionable, particularly considering the presence of cerebrospinal fluid. Additionally, there is also a disturbing lack of precision in identifying SCEA positioning in MRI scans, an issue that resonates with my critique about nitinol unfolding. Lastly, the authors' method for analyzing histology data is questionable, both in terms of visual interpretation and quantification of astrocyte activation intensity. The use of mean gray value as a threshold could potentially suppress actual signals, leading to a misrepresentation of astrocyte activation levels. Taken together, these issues seriously undermine the scientific integrity of the manuscript and therefore, it cannot be endorsed for publication in its current form. My detailed, point-by-point responses to the authors' rebuttal are provided below.

Response: We would like to express our gratitude to the reviewer for the meticulous evaluation of our revised manuscript. We deeply appreciate the valuable comments provided, as they have played a vital role in enhancing the quality of our research. We have given utmost consideration to these comments and have conducted additional experiments, performed further data analysis, and thoroughly revised the manuscript to address each concern raised by the reviewer. We are delighted to report that these revisions have significantly improved the overall quality of our work. We have provided detailed responses to each specific point raised by the reviewer in the following section.

Response to detailed comments:

1. My primary critique of this research paper, as I have formerly expressed, pertains to the **uncontrollable unfolding process of nitinol in an intracranial space** that cannot be visualized easily (e.g. optical microscopy) during delivery. The authors have yet to adequately address this critique in their response letter, as will be comprehensively elucidated in the following text.

In response to this critique, the authors have provided supplementary materials - namely Figure S2 and an accompanying video - in an effort to elucidate the unfolding process. This initiative is indeed appreciated. Nonetheless, it is imperative to underscore that these materials were obtained under *in vitro* conditions, as admitted by the authors. Consequently, it remains an open question as to whether this unfolding process, while meticulously visualized *in vitro*, would proceed identically within the intracranial space *in vivo*.

Response: We would like to express our gratitude to the reviewer for bringing up this important question. We understand your concerns regarding the direct visualization of the electrode deployment *in vivo*, considering the distinct conditions within the confined intracranial space under the cranium as compared to the *in vitro* environment. In order to directly observe the entire implantation process beneath the cranium, we thinned the rat skull by drilling, thereby allowing us to visualize the SCEA devices beneath the skull. The results are included as Supplementary Video 2, which provides a comprehensive overview of the implantation procedure *in vivo*. This video showcases the insertion of the SCEA strip, the expansion of the nitinol and electrode array beneath the skull, the retrieval of the nitinol actuator, and finally the removal of the electrode array from the brain surface through the cranial opening.

The video effectively demonstrates the reliable deployment of the electrode array beneath the cranium in live animals, illustrating the absence of any bleeding or damage throughout the process. These findings are consistent with the intact brain surface observed after removing the electrode array, as depicted in Figure 1k, as well as the MRI results shown in Figure 2.

Significantly, it is important to note that the deployed electrode array can be safely and completely removed from the brain surface through the small cranial opening, as demonstrated in the aforementioned video. This successful removal is attributed to the compliant nature and exceptional mechanical strength of the electrode array film. We believe that these results

effectively address the reviewer's concerns regarding the intra-operative application of the SCEA devices, as detailed below.

The authors have also supplemented the manuscript with Figure S7, which depicts the deployment of a SCEA within a beagle's brain. Yet, the description provided for this figure leaves a considerable degree of ambiguity regarding whether this deployment was executed within the intact intracranial space, or post-craniotomy. Inferred from the caption of Fig. 5b ("The craniotomy caused some bleeding, which left some blood on the surface of the dura mater"), the latter seems to be the case, which, if true, would mean the images captured post-craniotomy are of limited utility in validating the nitinol unfolding process within an intact intracranial space. The confines of the intracranial space beneath an intact cranium inherently impose limitations on the unfolding of the SCEA post-delivery. Conversely, these limitations are evidently absent when a craniotomy is performed.

Response: We apologize for any confusion related with Figure S7. In our study, for large animal models like beagle dogs, the subdural implantation of the SCEAs was conducted without durotomy, but it involved a craniotomy. Specifically, after the skull was removed, the SCEAs in the compressed state were inserted into the subdural space through a ~6 mm long dural slit without removing any dura mater. The SCEAs then expanded beneath the dura, as illustrated in Figure S7. Finally, the nitinol actuator was retracted through the dural slit.

This design choice was made because the standard approach of utilizing durotomy for subdural electrode array implantation presents higher risks than craniotomy. Removing a large-size dura mater can lead to changes in intracranial pressure, an increased risk of cerebrospinal fluid (CSF) leakage, and potential infections. To provide additional information, we have included the details of the above procedure on pages 10 and 21 of our manuscript. We hope these clarifications address any concerns and improve the understanding of our methodology.

In my initial review, I had assumed that the MRI images in Figure 2 were indicative of the SCEA's location. However, the authors' response indicates that the depiction of the SCEA's location was inferred rather than directly imaged - an approach that introduces subjectivity and detracts from the scientific rigor expected in such illustrations.

In summary, the authors' attempts to address my primary critique have not succeeded in presenting conclusive evidence validating the controllability of the nitinol unfolding process within an

intracranial space. The data provided either fall under *in vitro* studies or *in vivo* studies conducted post-craniotomy, both of which do not faithfully replicate the true conditions under a mostly intact skull aiming for minimal invasiveness. In light of the limitations posed by MRI in providing direct imaging of SCEA, owing to the presence of metals, I would recommend the authors consider leveraging high-resolution uCT techniques for *in situ* visualization of the unfolding process. It is critical that such data be included to comprehensively substantiate the unfolding process, which forms the backbone of this research paper and influences all subsequent data and claims.

Response: We would like to express our gratitude to the reviewer for the thorough evaluation of our manuscript and the valuable suggestion regarding *in vivo* observation of the device deployment process. However, due to the small thickness of the metal components in our device (only tens of nanometers), it is not feasible to achieve sufficient contrast for visualization using micro-computed tomography (μ CT).

Instead, as previously discussed, we employed an alternative approach for direct visualization of the entire epidural implantation process in rats *in vivo* by thinning the rat skull through drilling. This technique allowed us to capture the deployment process in real-time and *in vivo*. The accompanying video, titled "Supplementary Video 2," has been included in the revised version of the manuscript. Additionally, Figure S7 illustrates the deployment of a SCEA beneath the dura mater in a beagle dog *in vivo*.

We firmly believe that these results provide conclusive evidence regarding the reliability and safety of the SCEA deployment process within a confined space under the skull or beneath the dura mater. We appreciate the opportunity to address the reviewer's suggestion and provide further clarification on the deployment process.

2. In the authors' reply concerning the matter of epidural versus subdural delivery of the SCEA, **they have resorted to some factual inaccuracies to validate their implantation methodology.** To be specific, the authors argue that "*Subdural implantation can cause temporary damage to the arachnoid layer, which is a common challenge for all subdural recording methods.*" However, this assertion is fundamentally inaccurate.

Typically, ECoG arrays, especially in human applications, are directly placed on the leptomeninges surface, where the leptomeninges encompass both the arachnoid and pia mater. Several publications, including *Biomedical Microdevices*, 13, 59–68, 2011, have demonstrated that

conventional ECoG arrays do not inflict acute damage to the leptomeninges, although long-term deployment of ECoG arrays may induce leptomeninges thickening. Another study (Sensors 2021, 21, 178) indicated that the arachnoid membrane exhibits local depression due to the physical presence of grid electrodes while in the subdural space, but the arachnoid membrane is not ruptured (“While in the subdural space, tissue sections showed that arachnoid membrane was locally depressed by the occupying effect of grid electrodes.”).

Should the SCEA insertion and unfolding consistently inflict damage on the arachnoid layer, it points towards two unignorable facts: 1) The SCEA device, when in operation, incurs greater harm than a conventional ECoG device, and 2) The SCEA device’s unfolding process within the subdural space lacks sufficient control (thus confirming my previous comment), resulting in the rupture of the leptomeninges and blood vessels.

Response: We sincerely appreciate the detailed and valuable comments provided by the reviewer. We are pleased to provide a more comprehensive explanation regarding the potential damage to the arachnoid layer resulting from the subdural implantation of SCEAs and the conventional implantation of subdural electrode arrays through durotomy.

In careful examination of the surrounding meningeal structure of the brain in literature, it is noticed that the arachnoid mater consists of two distinct layers: the outer layer of arachnoid barrier cells and the inner layer of arachnoid trabeculae. It is worth noting that the arachnoid barrier cell layer is situated beneath the dura mater and exhibits a stronger attachment to the dura compared to the rest of the arachnoid, making its separation from the dura quite challenging. Conversely, the arachnoid trabeculae layer is characterized by web-like structures that traverse the subarachnoid space and have a less densely arranged configuration compared to the arachnoid barrier cell layer (Response Figure R1)¹

Response Figure R1¹. Diagram of the meningeal layers.

Considering the stronger attachment of the arachnoid barrier cell layer to the dura mater and the relatively loose packing of the arachnoid trabeculae layer, we reasonably suspect that the conventional implantation of subdural electrode arrays through durotomy would inevitably result in the separation of the arachnoid barrier cell layer from the rest of the arachnoid mater, thus disrupting the integrity of the meningeal layers. The schematic below from literature [2] also indicates that the subdural electrodes are located under the arachnoid, instead of above (Response Figure R2).

Response Figure R2². A depiction of surface brain layers showing subdural ECoG electrode placement, which is in the subarachnoid space.

In the case of the subdural implantation of SCEAs, the process involved the insertion of SCEA strips and the subsequent expansion of the SCEAs beneath the dura mater. Again, due to the stronger attachment of the arachnoid barrier cell layer to the dura mater, we reasonably suspect that the insertion and expansion of the SCEAs also occurred within the arachnoid mater, leading to the separation of the arachnoid barrier cell layer from the rest of the arachnoid mater.

We would like to emphasize that while it is important to investigate the impact of the implantation process on the meningeal layer, it is a relatively minor aspect of our studies. Our structural MRI studies on rats indicated no changes in brain structure or shape following SCEA implantation. DCE-MRI studies revealed no obvious breach of the cortical blood-brain barrier after SCEA implantation. Furthermore, histology studies indicated that the neurons in the cortex of rats showed a normal distribution with no noticeable tissue damage. And although the SCEA implants elicited a slight inflammatory response in acute stage, this inflammatory response diminished quickly over time. All of these results indicate that the implantation process, including SCEA strip insertion, deployment, and shape actuator retraction, was minimally invasive, and the SCEAs demonstrated excellent chronic biocompatibility. Furthermore, our devices successfully recorded high-quality electrocorticography (ECoG) signals both epidurally from rats and subdurally from beagles.

Therefore, whether the deployment of SCEAs resulted in a disruption of the meningeal layer is not a critical issue related to the application of SCEAs. We believe that our findings provide strong evidence of the reliability and safety of SCEAs, demonstrating their potential for practical applications.

References and citations supporting the above speculation are included as followings:

“The dura mater is composed of 2 layers: skull periosteum and meningeal layer. Internal to the meningeal layer is a unique layer of fibroblasts that has been termed the dural border cell layer. It is continuous externally with the meningeal dura and internally with the arachnoid, with no intervening subdural space. Thus, the term” subdural” when used in reference to the location of a fluid collection or hematoma is controversial as there is no evidence of a real anatomical space at this location.”¹

*“The arachnoid mater is an avascular meningeal layer. It is formed from 2 distinct cell layers. Adjacent to the dural border cells of the dura is the **arachnoid barrier cell layer**. This layer is full of densely packed cells joined together with numerous desmosomes and tight junctions. These provide the layer with a barrier function that prevents the movement of fluid across it. Deep to the arachnoid barrier cell layer is a more loosely packed layer of cells or **“arachnoid trabeculae.”** The cells bridge the subarachnoid space and attach to the pia as well as each other and also enclose vessels that traverse the layer.”¹*

“The arachnoid matter is the middle layer of the meninges. It contains the subarachnoid space which is filled with cerebrospinal fluid (CSF). The depth of the subarachnoid space is variable depending on the relationship between the arachnoid and pia mater.”¹

*“The pia is the innermost meningeal layer. It is a delicate, highly vascular, connective tissue envelope surrounding the brain and spinal cord. It forms a continuous layer of cells closely adherent to the brain surface that dip into the sulci and fissures. **From the brain surface, the pia is reflected onto blood vessels in the subarachnoid space to form their outer coating. This ensures separation of the subarachnoid space from the subpial and perivascular spaces of the brain)**”¹*

*“The arachnoid mater is a thin, translucent membrane located deep in the dura mater that contains a few layers of flattened cells, **under which the cerebrospinal fluid (CSF) flows in the subarachnoid space (SAS)**. This space contains trabeculae and collagen bundles generated by fibroblast-like cells that connect the arachnoid to the pia mater.”³*

*“The meninges comprise the dura mater and the leptomeninges (arachnoid and pia mater). Dura forms an outer endosteal layer related to the bones of the skull and spine and an inner layer closely applied to the arachnoid mater. Leptomeninges have multiple functions and anatomical relationships. **The outer parietal layer of arachnoid is impermeable to CSF due to tight intercellular junctions;** elsewhere leptomeningeal cells form desmosomes and gap junctions. Trabeculae of leptomeninges compartmentalize the subarachnoid space and join the pia to arachnoid mater.”⁴*

“The most superficial layer of the arachnoid, referred to as the subdural mesothelium, subdural neurothelium, or dural border layer, comprises layers of thin, densely arranged cells that abut the dura mater and is considered by Schachenmayr and Friede to be a portion of the dura. Adjacent

to this dural border layer is the arachnoid barrier layer, which consists of tightly packed polygonal cells, round nuclei coupled with pale cytoplasm, and a basement lamina that distinguishes it from the rest of the arachnoid. These cells are conjoined by characteristic tight junctions, absent in the dural border and desmosomes that form an impermeable barrier to CSF.”⁵

“The dividing line between the dura and the arachnoid did not impress one as a clear-cut boundary at which structure changed abruptly from one tissue compartment to the other. Instead, the uppermost layer of the arachnoid, its barrier layer, and the innermost portion of the dura, the dural border layer, were fused to form a layer of densely packed cells in which the attachment of cells to each other was much tighter than either that of the barrier layer to the arachnoid trabecules or that of the dural border cells to the remainder of the dura.”⁶

“The middle meningeal layer is the arachnoid mater, named for its peculiar spiderweb-like appearance. The arachnoid mater is composed of an outer layer of cells connected by tight junctions enclosing the subarachnoid space, which is filled with CSF and forms an effective physical barrier to the interstitial fluid (ISF) in the dura mater. Spanning the subarachnoid space are thin projections onto the pia mater called trabeculae, and major arteries penetrate the brain parenchyma from this CSF-filled cavity.”⁷

The invasive impact of ECoG is exemplified in the authors’ own data of Fig. 5b, where substantial bleeding is observed post nitinol retraction. For better clarity, I have highlighted these newly developed hemorrhagic spots with white arrows below. Given the gravity of these potential damage to the brain, they warrant immediate remediation before the publication of this method.

Response: We express our gratitude to the reviewer for raising this important issue and apologize for the confusion related to Figure 5b. The hemorrhagic spots, which were marked by the reviewer with white arrows in our figure, were actually from bleeding of the surrounding skull, rather than bleeding under the dura. The craniotomy procedure involved cutting through the skull and the blood vessels within, which inevitably led to bleeding. During the deployment process, the surrounding skull continued to bleed, resulting in the appearance of hemorrhagic spots on the surface of the dura.

As depicted in Figure 5b, the blood and blood vessels underneath the skull, with some examples marked by yellow arrows, displayed a much darker red color compared to the blood spots on the

dura surface, marked by white arrows below. The newly formed hemorrhagic spots depicted in the right image of Figure 5b, as indicated by the reviewer, actually exhibited a bright red color, indicating their location on top of the dura instead of beneath it.

We appreciate the reviewer's careful observation in pointing out this issue. We have added some discussion in the caption of Fig. 5b to make this issue clear. We hope that this clarification provides a more accurate understanding of the location and nature of the observed hemorrhagic spots.

Fig. 5b, Optical images acquired during the minimally invasive subdural implantation of a SCEA into the brain of a beagle dog before and after withdrawal of the nitinol shape actuator. The dura slit is marked by the white dotted line. The craniotomy caused some bleeding in the skull, leading to the formation of several hemorrhagic spots on the surface of the dura (white arrows), while the blood vessels underneath the dura displayed a much darker red colour (yellow arrow). The vessels under the dura remained intact, and no bleeding was observed during the SCEA insertion, deployment or shape actuator withdrawal processes. Scale bar, 5 mm.

3. In their responses to my comments regarding how the SCEA conforms to the cortical surface curvature, the authors referenced some facts that appear to be either irrelevant or inaccurately represented.

They state that "Studies have reported that mannitol can lead to a reduction of approximately 0.8% in the size of the human brain⁴. This reduction in brain volume creates sufficient space for the deployment of SCEAs without causing significant compression or displacement of brain tissue." Upon reviewing the referenced source (reference 4), it became clear that the 0.8% reduction pertains to the overall brain volume. From this data, one can conduct a basic mathematical analysis to determine the magnitude of size reduction across each dimension:

$$1-(1-0.8\%)^{1/3} = 0.27\%$$

Assuming a dorsoventral dimension of 1 cm for a mouse brain, this shrinkage corresponds to the creation of an additional intracranial space amounting to $1 \text{ cm} \times 0.27\% = 27 \text{ }\mu\text{m}$. Even when contrasted with the diameter of nitinol, which is 100 μm , this added space is relatively insignificant. Hence, **it is not scientifically sound to use this minuscule reduction in brain volume as a justification for facilitating the creation of space necessary for intracranial unfolding.**

Response: Again, we sincerely appreciate the thorough review conducted by the reviewer. With the addition of Supplementary Video 2, we were able to provide visual evidence that the insertion of SCEA strips, deployment of SCEAs, and retrieval of the nitinol actuators can all be smoothly and reliably performed beneath the skull. This indicates that the space between the skull and brain is indeed adequate to accommodate these processes.

We fully concur with the reviewer's calculations, which demonstrated that if there is a 0.8% reduction in brain size due to the application of mannitol, it would only result in a slight increase in the space between the skull and brain. Therefore, the natural space between the skull and brain must be a significant factor contributing to the successful implantation and deployment of SCEAs.

To gain some insight into the size of the natural space between the skull and brain, we consulted the mouse brain atlas (Response Figure R3, <http://labs.gaidi.ca/mouse-brain-atlas/>). Considering a coronal section 0.02 mm from bregma as an example, with the zero point on the dorsal-ventral axis marking the position of the skull surface, it can be observed that the distance between the outer surface of the skull and the brain surface is approximately 700 μm . Based on the literature⁸, the thickness of the skull ranges from ~100-600 μm , with the majority falling within 200 to 300 μm (Response Figure R4). Using these measurements, the space between the skull and brain surface in mice is estimated to be around 400 to 500 μm .

Response Figure R3. Mouse brain atlas (<http://labs.gaidi.ca/mouse-brain-atlas/>)

Response Figure R4⁸. Histograms of skull thickness measured from micro-CT scans of three mice. 64, 59 and 64 measurements were taken across the skull covering most of the dorsal cortex.

For the human brain, we researched the literature and discovered a paper that utilized MRI to study the space between the skull and brain surface⁹. The study provided clear imaging for measuring the scalp-brain distance. As illustrated in Response Figure R5, landmark A represented the inner margin of the cranial bone marrow, while landmark L represented the outer boundary of the cerebral cortex. The space between L and A encompasses the meninges, cerebrospinal fluid (CSF), and the inner table of the cranium. On T1-weighted MRI, the two contiguous compartments of CSF and the inner table cannot be differentiated since both appear dark. The L-A intervals exhibited a significant increase with age, with the distance expanding from 3.4 ± 1.9 mm in newborns to 7.0 ± 2.7 mm in 12-year-old children.

Taking these references into account, we can confidently conclude that the natural space between the skull and brain surface, for both rodents and primates, is indeed sufficient to accommodate the deployment of SCEAs. This conclusion is consistently supported by the newly added Supplementary Video 2 in the revised manuscript.

Response Figure R5. Components of brain-scalp space. Enlarged axial slice through a child brain (age 12 years). Red letters show manually selected landmarks.⁹

In addition to the previously mentioned contention, the authors present another argument, invoking the principle of capillary forces, to elucidate how the SCEA can adhere conformally to the brain surface: “due to its ultrasmall thickness (4 μm) and mesh structure, the electrode array would conformally adhered to the surface of the brain, as shown by the picture in Fig. 1j and 5b. That is to say, the conformal interfacing between the electrode array and the brain surface was actually formed under the surface tension due to the ultrathin nature and mesh structure of the electrode array film.”

This assertion, too, is scientifically flawed. Surface tension typically arises at the air-liquid interface, where an imbalance between adhesive and cohesive forces results in a net force known as surface tension, as detailed in the book “Intermolecular and Surface Forces” by J. N. Israelachvili. The intracranial space is filled with cerebrospinal fluid (CSF). Under the assumption that only a minor incision is made in the cranium to allow for SCEA insertion, it is reasonable to predict that this small opening would not lead to a significant drainage of the CSF (if it does, this would be considered as a major drawback of this method and should thus be explicitly mentioned).

Consequently, **the deployed SCEA within the intracortical space would be exposed to CSF on both sides, counterbalancing the surface tension.** A straightforward analogy can illustrate this concept: while cling wrap can adhere to the surface of food items owing to capillary forces when in air, this adhesion does not occur when the entire system is submerged in water.

Response: We are sincerely grateful for the insightful comments provided by the reviewer. We agree with the reviewer's assessment regarding the surface tension.

Upon careful reconsideration, considering the presence of cerebrospinal fluid (CSF), we now believe that the electrode array film may not strongly adhere to the brain surface. As demonstrated in the newly added Supplementary Video 2, the deployed electrode array can be reliably removed from the brain surface after implantation. This observation aligns with the speculation that there might indeed be CSF between the electrode array implants and the brain surface.

We would like to emphasize that the design of the SCEAs incorporates a mesh structure, which has been found to facilitate the extrusion and circulation of CSF following implantation¹⁰.

In light of these findings, we have made the necessary revisions to the manuscript by removing the claim related to the conformal interfacing.

We truly appreciate the reviewer's insightful input, which has significantly improved the accuracy and clarity of our work.

4. The authors' arguments about removing the SCEA from the surface of the brain do not entirely address the complexities inherent to this issue.

First, the comparison between the removal of SCEA and the removal of a stent from a blood vessel does not take into account the vast differences in scale and complexity between these procedures. A stent, when placed in a blood vessel, encounters a relatively uniform and predictable structure. The blood vessel is typically cylindrical with a relatively consistent diameter, which significantly simplifies the process of stent removal. Conversely, the SCEA is implanted within the irregular, convoluted topography of the brain, with its myriad folds and crevices. Moreover, the brain's tissue is markedly more sensitive and prone to damage compared to the walls of a blood vessel. The sheer thinness and delicate mesh structure of the SCEA, as acknowledged by the authors, make it extremely vulnerable to breakage during removal. This could potentially result in fragments of the array being left behind in the brain, which poses serious risks and requires larger-

area craniotomy for retrieval. Furthermore, most stents are designed for permanent implantation, which eliminates the issue of removal altogether.

In addition, in light of this response and another response later to chronic studies, the authors seem to suggest the SCEA technology is primarily intended for single-time-point, intraoperative monitoring. If this is indeed the case, then it becomes imperative that the SCEA be removed post-operation to prevent any possible chronic damage - an aspect that has not been sufficiently addressed in the study. If the argument is that the subject animal would be sacrificed after the experiment, thereby negating the need for SCEA removal, it calls into question the necessity of minimally invasive implantation. If it is a terminal experiment, craniotomy could be performed, as this invasive procedure would not factor into the long-term welfare of the subject.

Response: We express our sincere appreciation for the valuable comments provided by the reviewer. We acknowledge the significance of ensuring the safe and reliable removal of the SCEAs in their practical application.

In support of this, we have included a newly added supplementary movie 2, which demonstrates the safe retraction of the SCEA through the small cranial opening used for the SCEA implantation. Notably, at the end of the video, the removed SCEA remains intact without any signs of breakage. We attribute this successful removability to the compliant nature and excellent mechanical strength of the thin electrode film.

We believe that the combination of the electrode film's compliant properties and its remarkable mechanical strength offers a guarantee for the SCEAs' safe removal.

Once again, we would like to express our gratitude to the reviewer for raising this important point, and we are pleased to address it in our revised manuscript.

5. The response by the authors regarding the positioning of the SCEA in MRI images raises some concerns about the precision and objectivity of this method. By stating that the SCEAs are “invisible” in the MRI scans and that their position was determined based on the “implantation position and the geometry of the array,” it appears that the positioning of the SCEA is marked based on **an assumption rather than precise, empirical, and objective data**.

This method of manually labeling the position of the SCEA seems to rely heavily on subjective judgement and is therefore prone to inconsistencies or errors, especially when different individuals

are involved in the process. It also suggests a lack of rigor, as subjective human estimations are inherently less reliable and less precise than measurements made with objective, scientifically verifiable methods. This lack of rigor is consistent with my first critique above, highlighting the **uncontrollable unfolding process of nitinol in an intracranial space** that is not guided by intraoperative imaging but only relies on *ex vivo* experiments or *in vivo* experiments post-craniotomy.

The authors' reliance on an **approximation** of the SCEA's position in the MRI images does NOT align with best scientific practices, which emphasize precision, replicability, and objective measurements. Given the potential implications of inaccuracies in the SCEA positioning, it is critical that a more objective and quantifiable method of determining the SCEA position is adopted. A more reliable approach could involve using a contrast agent or other MRI-visible markers incorporated into the SCEA design that could be accurately detected in the MRI scans. Alternatively, as suggested above, the authors should consider leveraging high-resolution uCT techniques for *in situ* visualization of the metal components in SCEA under the intact cranium.

Response: We express our deep gratitude to the reviewer for raising this crucial concern. We fully recognize the importance of accurately localizing the electrode sites on the brain surface for the practical application of SCEAs. Since craniotomy or durotomy is not performed during the implantation process, intraoperative visualizations and photographs of the electrode contacts and cortical surfaces are not possible. To address this challenge, we have developed two methods: postoperative localization and intraoperative localization of the SCEAs.

For postoperative localization, we have utilized magnetic resonance imaging (MRI) of the SCEAs with a thin nickel layer incorporated as a contrast enhancer. This technique enables direct visualization of the implants and allows for the determination of the spatial positioning of the SCEAs on the brain surface after surgery. This method aligns closely with the suggestion made by the reviewer. Additionally, for intraoperative localization, we have leveraged the Brainsight™ neuronavigation system, which is a commercially available tool. Detailed procedures for both postoperative and intraoperative localization have been included in the "SCEA localization" sections of the results and the methods section in the revised manuscript.

Regarding the MRI studies presented in Figure 2, the localizations of the SCEAs were established based on the implantation position and the geometry of the SCEAs electrode. Through dynamic

contrast-enhanced MRI (DCE-MRI) examinations, we observed that the position of the continuous segments of convexity meningeal accurately corresponded with the determined location of the SCEA implants. This finding demonstrates the effectiveness and precision of our localization method. As the reviewer correctly pointed out, the prerequisite for this method is the well-controlled and complete deployment of the SCEAs. As discussed earlier and demonstrated in the newly added supplementary video showcasing the *in vivo* implantation and deployment of the SCEAs, we have successfully met this prerequisite in our study. We firmly believe that this approach ensures the reliability of the conclusions drawn from the MRI results.

By implementing these localization methods, we have aimed to overcome the limitations posed by the absence of intraoperative visualizations, thereby ensuring the robustness of our findings and their practical relevance.

Once again, we express our utmost appreciation to the reviewer for raising this important point, and we are pleased to have addressed it in our revised manuscript.

6. Regarding the analysis method used for histology data, I have attached the Week-4 GFAP data below and have the editor and other reviewers (maybe additional reviewers too) for everyone's judgment. The authors assert that the normalized signals for the implanted and control sections are 0.70 and 1.01, respectively, within the UL layer, thus indicating a higher GFAP signal in the control as compared to the implanted section. Contrary to the authors' interpretation, the image below clearly illustrates a more pronounced GFAP signal (in green) in the implanted section than in the control part.

I would also like to question the authors' analysis method for quantifying the intensity of astrocyte activation and accumulation. The authors state that "the intensity of astrocytes activation and accumulation was determined by calculating the proportion of the area occupied by GFAP signals above a threshold intensity level, which was the mean gray value of all the pixels in layer I, II, III and IV on each image. The pixels with intensity above the mean were defined as signals." Inherent to this methodology is **a systematic issue that may inadvertently skew the results**. Specifically, the use of a threshold intensity level based on the mean gray value might inadvertently suppress the actual signal, especially in images where GFAP signals are highly represented. Consequently, images with higher GFAP signal intensity would naturally yield a higher mean gray value, thereby elevating the threshold. In turn, this could result in fewer pixels being defined as signals,

potentially underestimating the true intensity of astrocyte activation and accumulation. This approach may thus obscure the actual extent of astrocyte activation and accumulation, leading to

potentially misleading conclusions. I believe this critical aspect warrants a more in-depth discussion and perhaps, a revision in the methodology.

Response: We express our sincerest gratitude to the reviewer for conducting a thorough examination of our manuscript and bringing to our attention the issues with our analysis of astrocyte intensity. We fully acknowledge the validity of the reviewer's concerns regarding the previous method, which may have resulted in systematic errors due to inappropriate calculation of threshold values. This discrepancy likely explains the inconsistency between the calculated astrocyte intensity values and the visual differences observed in the figures mentioned.

After careful consideration, we have made significant changes to our astrocyte activation analysis by revising the calculation of threshold values. Instead of using the mean gray value of pixels in layers I, II, III, and IV, we now calculate the threshold intensity using the mean gray value of pixels in layers V and VI for each image. This adjustment is based on the rationale that the SCEAs were implanted on the cortical surface, and the lower layers of the cortex are least affected by the SCEA implants. Importantly, we observed no significant difference in the threshold values between the control and implant groups when calculated using this revised method (Response Figure R6). This provides substantial validation for the accuracy of this analysis approach.

Using the new threshold values, we have recalculated the intensity of astrocyte activation and accumulation, and we have updated the results in Figure 3 of the revised manuscript. The new findings indicate that the GFAP level was elevated at the implanted side in the L1 layer at 1 week postimplantation, instead of having no difference from the control side concluded using the old analysis.

Addressing the specific concern raised by the reviewer regarding the data at 4 weeks postimplantation in the UL layer, we calculated the normalized signals for the implanted and control sides using the new analysis method, resulting in values of 1.04 and 1.00, respectively. Statistical analysis of all samples revealed no significant difference between the two groups (new Figure 3e).

Once again, we extend our utmost appreciation to the reviewer for bringing the issue with astrocyte intensity to our attention. We are pleased to have addressed it by implementing a more accurate analysis method in our revised manuscript. We firmly believe that this revision significantly enhances the scientific rigor of our research findings.

Response Figure R6. Mean grayscale intensity of GFAP in V and VI layers. Individual data points are overlaid on the box plots. The box plots show the median and quartile range, and the whiskers denote $1.5 \times$ the interquartile range. N=16 from 4 rats for weeks 1 and 2 data, n=29 from 4 rats for week 4 data, n=23 from 6 rats for week 8 data. For the data that could be represented by a normal distribution, paired t-tests were used. Otherwise, Wilcoxon signed-rank tests were used in the significance analysis. ns, $p > 0.05$, *, $p < 0.05$, **, $p < 0.01$, ***, $p < 0.001$.

Figure 3. d-e, Normalized signal of GFAP-labelled astrocytes in layer I (d) and the upper layers (e).

Reference:

1. Patel, N. & Kirmi, O. Anatomy and imaging of the normal meninges. in *Seminars in Ultrasound, CT and MRI* vol. 30 559–564 (Elsevier, 2009).
2. Obidin, N., Tasnim, F. & Dagdeviren, C. The future of neuroimplantable devices: a materials science and regulatory perspective. *Adv. Mater.* **32**, 1901482 (2020).
3. Ma, T., Wang, F., Xu, S. & Huang, J. H. Meningeal immunity: Structure, function and a potential therapeutic target of neurodegenerative diseases. *Brain. Behav. Immun.* **93**, 264–276 (2021).
4. Weller, R. O. Microscopic morphology and histology of the human meninges. *Morphologie* **89**, 22–34 (2005).
5. Mortazavi, M. M. et al. Subarachnoid trabeculae: a comprehensive review of their embryology, histology, morphology, and surgical significance. *World Neurosurg.* **111**, 279–290 (2018).
6. Schachenmayr, W. & Friede, R. L. The origin of subdural neomembranes. I. Fine structure of the dura-arachnoid interface in man. *Am. J. Pathol.* **92**, 53 (1978).
7. Alves de Lima, K., Rustenhoven, J. & Kipnis, J. Meningeal immunity and its function in maintenance of the central nervous system in health and disease. *Annu. Rev. Immunol.* **38**,

597–620 (2020).

8. Ghanbari, L. et al. Craniobot: A computer numerical controlled robot for cranial microsurgeries. *Sci. Rep.* **9**, 1023 (2019).
9. Beauchamp, M. S. et al. The developmental trajectory of brain-scalp distance from birth through childhood: implications for functional neuroimaging. *PLoS One* **6**, e24981 (2011).
10. Branco, M. P., Geukes, S. H., Aarnoutse, E. J., Ramsey, N. F. & Vansteensel, M. J. Nine decades of electrocorticography: A comparison between epidural and subdural recordings. *Eur. J. Neurosci.* **57**, 1260–1288 (2023).

REVIEWERS' COMMENTS

Reviewer #2 (Remarks to the Author):

The authors have diligently addressed my previous round of comments through thorough analysis and additional experiments. All my concerns have been resolved.